# An allosteric ligand-binding site in the extracellular cap of K2P channels

Qichao Luo[1,2], Liping Chen[1,2], Xi Cheng[1,2], Yuqin Ma[1,2], Xiaona Li[1,2], Bing Zhang[1,2], Li Li[1,2], Shilei Zhang[3], Fei Guo[1,2], Yang Li [1,2] & Huaiyu Yang[1,2,4]

Two-pore domain potassium (K2P) channels generate leak currents that are responsible for the maintenance of the resting membrane potential, and they are thus potential drug targets for treating diseases. Here, we identify N-(4-cholorphenyl)-N-(2-(3,4-dihydrosioquinolin-2 (1H)-yl)-2-oxoethyl)methanesulfonamide (TKDC) as an inhibitor of the TREK subfamily, including TREK-1, TREK-2 and TRAAK channels. Using TKDC as a chemical probe, a study combining computations, mutagenesis and electrophysiology reveals a K2P allosteric ligand-binding site located in the extracellular cap of the channels. Molecular dynamics simulations suggest that ligand-induced allosteric conformational transitions lead to blockage of the ion conductive pathway. Using virtual screening approach, we identify other inhibitors targeting the extracellular allosteric ligand-binding site of these channels. Overall, our results suggest that the allosteric site at the extracellular cap of the K2P channels might be a promising drug target for these membrane proteins.

---

[1] State Key Laboratory of Drug Research and Key Laboratory of Receptor Research, Shanghai Institute of Materia Medica, Chinese Academy of Sciences, 555 Zuchongzhi Road, Shanghai 201203, China. [2] University of Chinese Academy of Sciences, No. 19A Yuquan Road, Beijing 100049, China. [3] Jiangsu Key Laboratory of Translational Research and Therapy for Neuro-Psycho-Diseases & Department of Medicinal Chemistry, College of Pharmaceutical Sciences, Soochow University, 199 Ren'ai Road, Suzhou 215123, China. [4] Shanghai Key Laboratory of Regulatory Biology, Institute of Biomedical Sciences and School of Life Sciences, East China Normal University, 500 Dongchuan Road, Shanghai 200241, China. Qichao Luo, Liping Chen, Xi Cheng and Yuqin Ma contributed equally to this work. Correspondence and requests for materials should be addressed to Y.L. (email: liyang@simm.ac.cn) or to H.Y. (email: hyyang@bio.ecnu.edu.cn)

Fifteen two-pore domain potassium (K2P) channels have been identified in the human genome[1, 2]. They contribute to the background leak currents responsible for the maintenance of the resting membrane potential. Linked to several pathologies, K2P channels represent important clinical targets in the treatment of cardiovascular disease and neurological disorders, including pain and depression[3]. For example, the TREK-1 channel contributes to the perception of pain, regulation of mood, anesthetic responses, cardiac mechanoelectric feedback and vasodilation[4–9] and is involved in the glutamate conductance and the regulation of blood–brain-barrier permeability[10–12]. Therefore, modulators targeting K2P channels would be therapeutically useful for the design of drugs treating relevant diseases. To progress toward a successful rational drug design targeting K2P channels, a basic understanding of how ligands interact with these proteins is necessary.

The currently available crystal structures of K2P channels have revealed information about how these channels respond to ligands. In these structures, K2P channels are homogenous dimers. Each monomer includes two extracellular helices (E1 and E2), two-pore domains (P1 and P2), and four transmembrane helices (M1-M4)[13–18]. In the transmembrane domain formed by the M2-M4 helices, there are prominent fenestrations connecting the inner pore with the milieu of the membrane. These fenestrations could be occupied by lipid acyl chains or small molecular ligands that project into the intracellular ion conducting pore, thus contributing to a non-conductive channel[15, 16, 18].

A rather unique structural feature of K2P channels is the extracellular cap formed by the E1 and E2 helices, which is not observed in other ion channels. In some K2P channels, an apical disulfide bridge stabilizes the E1 and E2 helices[19–21]. This extracellular domain defines two tunnel-like side portals as the extracellular ion pathway and partially obstructs the direct movement of ions into the extracellular milieu[22–25]. Compared with classical potassium channels, K2P channels offer bilateral extracellular access to the selectivity filter. This distinguishing extracellular ion pathway explains the insensitivity of K2P channels to the classical potassium channel pore blockers, such as tetraethylammonium, 4-aminopyridine, and cesium ion[26, 27].

In this study, we find that through interactions with the extracellular cap, N-(4-cholorphenyl)-N-(2-(3,4-dihydrosioquinolin-2 (1H)-yl)-2-oxoethyl)methanesulfonamide (TKDC, Fig. 1a) is able to inhibit all three members of the TREK subfamily (TREK-1, TREK-2 and TRAAK). Using computational modeling, mutagenesis, and electrophysiology with chemical probes, we characterize the binding mode of TKDC to TREK-1 and provide a molecular explanation for the TKDC-induced allosteric conformational transitions. We discover more inhibitors by applying virtual screening to this binding site, which further supports the idea that the extracellular cap of K2P channels is a functionally important drug target. Our results suggest that the allosteric conformational transitions induced by the interaction of inhibitors with the extracellular cap of K2P channels may provide a molecular basis for the development of drugs targeting K2P channels.

## Results
**Inhibition of TREK channels by TKDC.** In the experimental screening, we identified TKDC as an inhibitor of TREK-1 from an in-house library of approximately 1000 small molecules developed over time. In the whole-cell voltage clamp experiments on CHO cells that were transiently transfected with TREK-1, pronounced decreases in current were generated by perfusion with 10 μM TKDC (Fig. 1b, c). The inhibitory effect of TKDC on the TREK-1 channel was dose-dependent with a half-maximal inhibition concentration (IC$_{50}$) of $4.9 \pm 0.6$ μM (Fig. 1d, e).

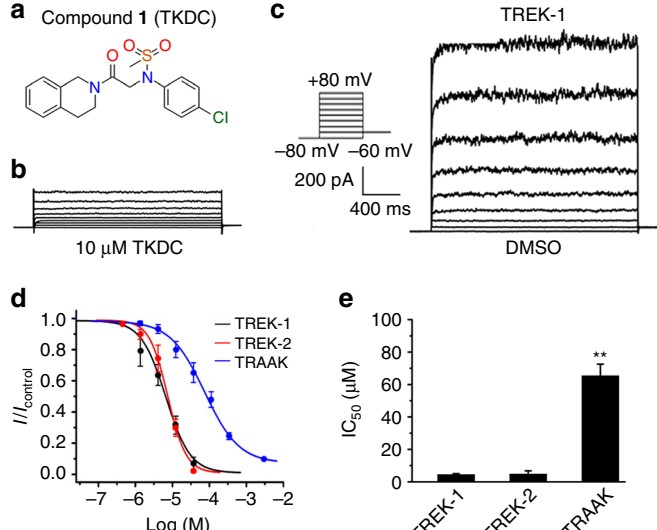

**Fig. 1** Inhibition of TREK subfamily channels by TKDC in CHO cells. **a** Chemical structure of TKDC. **b, c** Typical whole-cell current traces recorded from CHO cells overexpressing the TREK-1 channel with 10 μM TKDC **b** or DMSO application **c**. Currents were elicited by depolarizing voltage steps from a holding potential of −80 mV to + 80 mV in 20 mV increments, followed by stepping down to −60 mV. **d** Dose-dependent inhibition of TKDC on TREK-1, TREK-2 and TRAAK channels. **e** The statistics of the half-inhibitory concentrations of TKDC for TREK-1 ($n = 7$), TREK-2 ($n = 7$), and TRAAK ($n = 8$) channels. IC$_{50}$ values were obtained via dose–response fitting. The Kruskal–Wallis test was used for statistical analysis; ** indicates $P < 0.01$. The data are shown as the mean ± s.e.m

We then investigated whether other TREK subfamily members were also inhibited by TKDC. During TKDC treatments, significant inhibition (IC$_{50}$: $5.2 \pm 1.9$ μM) was observed in CHO cells expressing TREK-2 (Fig. 1d, e). However, TKDC showed a weak inhibitory effect on TRAAK with an IC$_{50}$ value of $65.9 \pm 7.0$ μM, which is ~11-fold higher than that for the inhibition of the TREK-1 channel (Fig. 1d, e). These results suggest that TKDC is a common inhibitor for all three TREK channels with different potencies.

In the electrophysiological experiments investigating TREK-1, the incubation of cells with 10 μM TKDC significantly decreased the current, which suggested the possibility of the internalization of TREK-1. Therefore, we used confocal microscopy to investigate the co-localization of TREK-1 with Rab5a (an early endosome marker protein) under the same conditions. Compared with the DMSO control, TKDC did not significantly increase TREK-1 internalization (Supplementary Fig. 1a, b). Then, we used flow cytometry to examine TREK-1 cell membrane expression. TREK-1 positive cells did not obviously lose the cell surface signal after TKDC incubation (Supplementary Fig. 1c). Both results exclude the possibility that the TKDC-induced endocytosis of TREK-1 yielded the significant decrease in channel current observed in the electrophysiological experiments.

To further validate the direct binding of TKDC to the TREK-1 channel, we tested the TKDC-induced inhibition of TREK-1 in inside-out and outside-out patches. The application of TKDC to the extracellular face of the membrane (outside-out recording) consistently inhibited TREK-1 (Supplementary Fig. 2a). In contrast, the application of TKDC to the intracellular side of the channel did not significantly inhibit TREK-1 activity (Supplementary Fig. 2b). These results provide evidence that TKDC directly inhibits TREK-1 and the inhibition requires access to the extracellular cap domain.

**Binding mode of TKDC to TREK-1.** To investigate the possible binding modes of TKDC to the TREK channels, we adopted a two-stage computational approach. In the first stage, molecular docking methods (Schrodinger Glide[28] and Autodock[29]) were applied to the TREK family crystal structures to identify possible binding site(s) of TKDC across the entire family. We located 36 potential TKDC-binding sites (termed sites 1–36) with low docking scores (Glide G-scores < −3.0 or Autodock binding score < −6.0) in the twelve crystal structures of TREK channels (Supplementary Fig. 3). Some of these predicted binding sites were equivalent in the different crystal structures. For example, sites 1, 2, 4–8, 11 and 21 were all binding pockets consisting of the M1, M2, M4 and P1 helices and shared conserved residues (Supplementary Fig. 4a–f). Thus, according to the locations and compositions of residues, we assigned all 36 predicted binding sites to 11 groups (groups A–K). All sites were ranked based on the Glide G-score and Autodock binding score, respectively (Supplementary Fig. 3b). Regardless of docking methods (Glide and Autodock), the predicted binding sites in groups A and B always had the lowest docking scores (Glide G-scores < −7.5 or Autodock binding score < −9.0). In group A sites (sites 1, 2, 4–8, 11 and 21), TKDC was surrounded by hydrophobic residues, which provided an unfavorable environment for the charged sulfonyl group of TKDC (Supplementary Fig. 4a–f). Therefore, these sites may not bind to TKDC. In group B, there was only one member, i.e., site 3, which had the third lowest G-score and the lowest Autodock binding score. Site 3 was located in the extracellular cap domain consisting of two helices from different subunits in TREK-1 (Fig. 2a). This site was only observed in a crystal structure of TREK-1 (Protein Data Bank (PDB) code 4TWK). No inhibitors of K2P channels have been reported to bind to site 3. In site 3, TKDC showed a more reasonable binding pose with its hydrophobic groups inserted into a gap between the extracellular helices and the polar sulfonyl group exposed to the extracellular solvent (Supplementary Fig. 4g, h). Thus, site 3 was considered as the possible binding site of TKDC in TREK-1 and was selected for further analysis. Notably, site 3 is an extracellular site, which is consistent with the results of the inside-out and outside-out patch experiments. In the second stage, considering the receptor flexibility and environmental effects, RosettaLigand[30–33] was applied to accurately characterize the docking mode of TKDC into the identified binding site, i.e., site 3. The top ten docking models with the lowest binding energy exhibited good structural convergence and were consistent with the first-stage docking results. In the docking model with the lowest binding energy (Fig. 2), the bicyclic group of TKDC was embedded in a deep cavity formed primarily by residues I80, Q83, L102 and Q105. Near the bottom of the extracellular cap domain, the chlorophenyl group of TKDC was positioned in a shallow cavity formed by Q76, T79 and I80. The binding pose of TKDC in TREK-2 was similar to that in TREK-1 (Supplementary Fig. 5).

The molecular docking of the TKDC/TREK-1 complex suggests that TKDC might bind to the extracellular cap of TREK-1. To verify this possibility, we mutated the residues in the putative TKDC-binding site and examined the inhibition of mutants by TKDC. Mutagenesis manipulation revealed that the kinetics of TREK-1 was intact (Supplementary Fig. 6a). Compared with wild-type (WT) TREK-1, the Q76A, I80A, and L102A mutants were less sensitive to TKDC (Fig. 2c and Supplementary Fig. 6c, d). The mutation of these residues into alanine significantly increased the $IC_{50}$ for TKDC-induced inhibition by at least -twofold (Fig. 2d). These results are consistent with our docking results and suggest that Q76, I80 and L102 are important residues for TKDC binding to the TREK-1 channel. The T79A, Q83A, and Q105A mutants showed no significant changes in TKDC-induced inhibition compared with

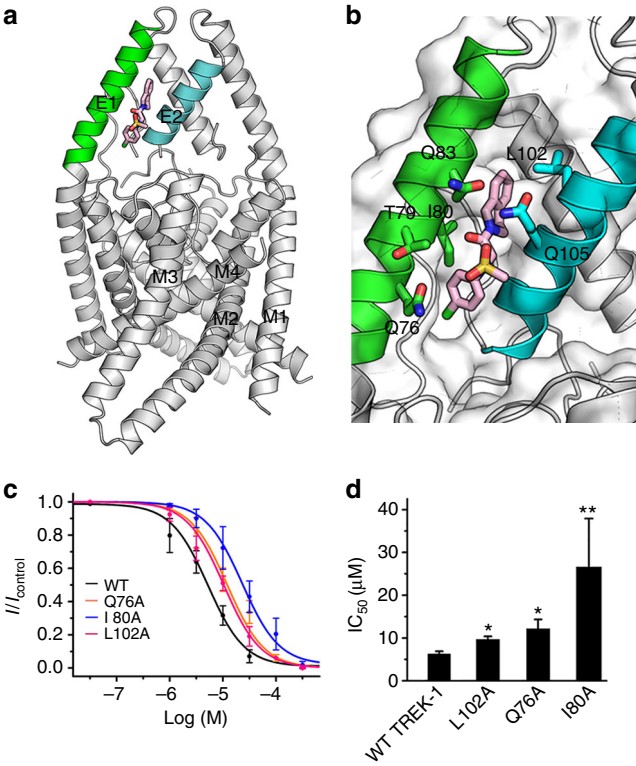

**Fig. 2** Binding mode of TKDC to TREK-1. **a** Binding site of TKDC in the extracellular cap of the TREK-1 channel. **b** Detailed view of the interactions between TKDC and the TREK-1 channel. TKDC and the residues involved in binding are shown as sticks. The protein is shown in a cartoon and surface depiction. **c** Dose-dependent inhibition of TKDC on the WT and mutant TREK-1 channels. **d** Histograms summarizing the half-inhibitory concentrations for WT ($n = 7$), L102A ($n = 10$), Q76A ($n = 6$) and I80A ($n = 3$) mutant TREK-1 channels. $IC_{50}$ values were obtained via dose–response fitting. The Kruskal–Wallis test was used for statistical analysis; * indicates $P < 0.05$, ** indicates $P < 0.01$. The data are shown as the mean ± s.e.m

the WT TREK-1 channel (Supplementary Table 1), indicating that the polar side chains of these residues were not critical for TKDC binding.

**Mutagenesis of TRAAK extracellular cap.** Compared with TREK-1 and TREK-2, TRAAK was less sensitive to TKDC. According to the multiple sequence alignment of the extracellular caps of TREK channels, TRAAK has three unique residues, A35, E38 and V42, which are situated in the binding site of the extracellular cap (Fig. 3a). We substituted these residues (A35, E38 and V42) of TRAAK with the corresponding residues (Q76, T79 and Q83, respectively) in TREK-1, and tested the TKDC-induced inhibition of these mutant TRAAK channels. As shown in Fig. 3b, c, the substitution of these residues resulted in enhanced inhibition of TRAAK by TKDC, indicating that TKDC bound to the proposed binding site in the extracellular cap of TRAAK.

The available crystal structures of TRAAK do not display the proposed extracellular binding site. To further investigate the structural basis for the enhanced TKDC-induced inhibition in mutants, we built homology models of three TRAAK mutants (A35Q, E38T and V42Q) using the TREK-1 crystal structure as the template and performed docking of TKDC to these mutant models. In the docking models of the A35Q mutant, the outward-facing Q35 with a larger side chain could provide a larger contact surface with TKDC than the alanine in WT TRAAK (Fig. 3d).

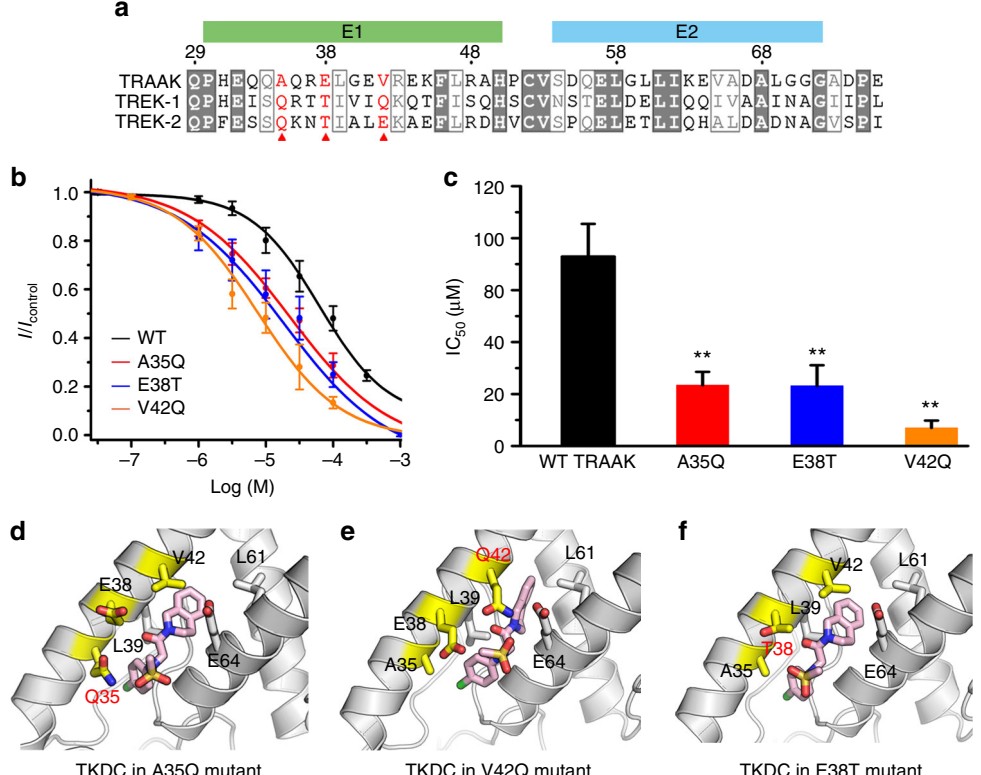

**Fig. 3** Substitutions of key residues in the extracellular cap of TRAAK channel. **a** Multiple sequence alignment for the extracellular caps of TREK-1, TREK-2, and TRAAK channels. **b** Dose-dependent inhibition of TKDC on the WT and mutant TRAAK channels. **c** Histograms summarizing the half-inhibitory concentrations of TKDC for the WT ($n = 8$), A35Q ($n = 8$), E38T ($n = 6$) and V42Q ($n = 7$) mutant TRAAK channels. Mutations of these residues in the extracellular cap showed enhanced inhibitory effects of TKDC. $IC_{50}$ values were obtained via dose–response fitting. One-way ANOVA with post hoc LSD test was used for statistical analysis [$F$ (3, 20) = 18.551]; ** indicates $P < 0.01$. The data are shown as the mean ± s.e.m. **d**, **e** Docking modes of TKDC in A35Q TRAAK **d**, in V42Q TRAAK **e**, and in E38T TRAAK **f**. The protein is shown as a cartoon. TKDC and key residues are shown as sticks

These interactions might help stabilize the insertion of TKDC in the extracellular cap. In the docking models of the V42Q mutant, in contrast to a valine interacting with nearby hydrophobic residues (e.g., L61), Q42 flipped outward to yield an expanded cavity between the E1 and E2 helices. As a result, the bicyclic hydrophobic group of TKDC could insert deeply into the channel cap, as shown in Fig. 3e. In the WT TRAAK, E38 was located in the center of the gap between the E1 and E2 helices. This negatively charged residue might repel TKDC, which contains a negatively charged sulfonyl group. As shown in Fig. 3d–f, TKDC and the side chains of the mutants were arranged to maximize the distance between E38 and the sulfonyl oxygens in both docking models. In the docking models of the E38T mutant, the negatively charged E38 was substituted with a polar threonine residue. Because of the reduced electronic repulsion in E38T, TKDC might have a higher affinity with the extracellular binding site. These electronic interactions could be critical in the binding of ligand to the K2P channel at the extracellular cap; this topic is discussed later.

**Ligand-binding mode of TRAAK examined by a TKDC derivative**. The docking models of A35Q and V42Q TRAAK suggest that ionic interactions between E38 and the sulfonyl group of TKDC might affect the binding of TKDC to TRAAK. We designed a ligand termed 28NH by excluding the sulfonyl group in TKDC (Fig. 4a). 28NH significantly inhibited the TRAAK channel with an $IC_{50}$ of $11.8 ± 2.1$ μM, which was almost sevenfold lower than the value obtained with TKDC (Fig. 4b, c). However, the $IC_{50}$ values for 28NH-induced inhibition of TREK-1 and TREK-2 were

comparable to those observed for TKDC-induced inhibitions (Fig. 4c). These observations indicate that the sulfonyl group makes a negative contribution to the binding of TKDC to TRAAK, but is acceptable for TREK-1 and TREK-2, both of which lack negatively charged residues in the center of the binding site. In the docking models of TRAAK/28NH, 28NH displayed a reasonable binding pose with its hydrophobic groups deeply inserted into the proposed extracellular site of TRAAK (Fig. 4d).

**Ligand-induced blockage of ion conductive pathway**. In the experiments mentioned above, we identified the binding site and characterized the binding mode of TKDC to TREK-1. With this knowledge, we aimed to simulate the process of the TKDC-induced inhibition of TREK-1 using molecular dynamics (MD) simulation approach. A TREK-1/TKDC complex simulation system (termed $S_{complex}$) was built by inserting the docking model of TREK-1/TKDC into an explicit membrane environment. To investigate the influence of TKDC, a system without TKDC (termed $S_{apo}$) was also established as a control. To build the $S_{apo}$ system, we removed TKDC from the docking model of TREK-1/TKDC and placed the protein in the same environment as that of the $S_{complex}$ system. Each system was replicated to perform two to three independent simulations. In the $S_{complex}$ simulations, the interactions between TKDC and TREK-1 were stable. The dominant pose of TKDC (Supplementary Fig. 7) was very similar to its starting pose (Fig. 2b). To further characterize the binding mode of TKDC to TREK-1, we calculated the frequency of interaction between a

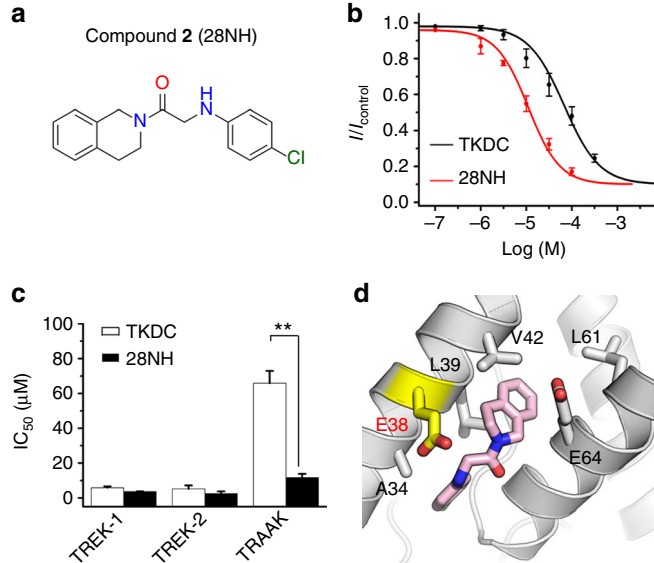

**Fig. 4** Inhibition of TREK subfamily channels by 28NH in CHO cells. **a** Chemical structure of 28NH. **b** Dose-dependent inhibition of TRAAK by TKDC and 28NH. **c** The statistics of $IC_{50}$ values for the inhibition of TREK-1, TREK-2 and TRAAK channels by TKDC and 28NH. $IC_{50}$ values were obtained via dose–response fitting. The unpaired t-test was used for statistical analysis [$t(7) = 1.027$ and $n = 9$ for TREK-1, $t(4) = 0.910$ and $n = 6$ for TREK-2, and $t(6) = 5.724$ and $n = 8$ for TRAAK]; ** indicates $P < 0.01$. The numbers in the bars indicate the number of cells studied per condition. The data are shown as the mean ± s.e.m. **d** Binding model of 28NH to TRAAK. The protein is shown as a cartoon. 28NH and key residues are shown as sticks

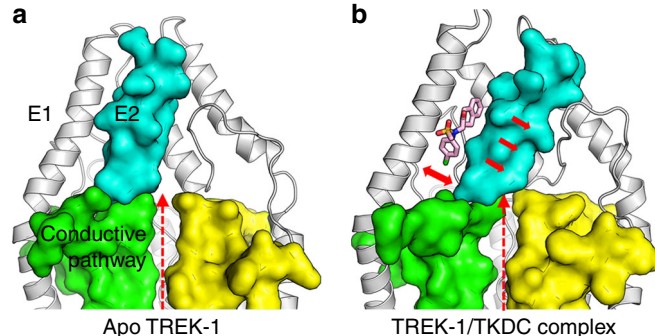

**Fig. 5** Different conformations of TREK-1 obtained from MD simulations. **a** Conformation of TREK-1 in the apo system $S_{apo}$. **b** Conformation of TREK-1 in the complex system $S_{complex}$ with TKDC. The protein is shown as a cartoon. The extracellular E2 helix (*cyan*) and the selectivity filter below (*green* and *yellow*) are shown as surfaces. The filter in green and the E2 helix in cyan are in the same subunit. The yellow filter is in the other subunit. TKDC is shown as sticks. The red dashed arrow represents the conductive pathway. The red solid arrows indicate the movement of the E2 helix under the influence of TKDC. All illustrations are from the end of simulation runs

particular residue and TKDC in the $S_{complex}$ system simulations. The 'interaction' is defined based on distance: when a residue has one or more atoms within 0.4 nm of TKDC, it interacts directly with TKDC. In the last 200 ns trajectory of each $S_{complex}$ system simulation, I80 and L102 interacted directly with the bicyclic group of TKDC at a high frequency (more than 0.85). Each of Q76, T79 and I80 interacted with the chlorophenyl group of TKDC at a frequency greater than 0.6. This result is highly consistent with our docking and mutagenesis studies.

In the last 200 ns simulations, the conformations of the TREK-1 channel in the system $S_{complex}$ were different from the conformations observed in the system $S_{apo}$ (Fig. 5). Compared with the apo system $S_{apo}$ without an inserted ligand, $S_{complex}$ had a wider gap between the E1 and E2 helices, especially at their bottoms near the selectivity filter (Supplementary Table 2). In $S_{complex}$, the average minimal distance between the bottom of the E2 helix (cyan in Fig. 5b) and the filter loop upstream pore helix P1 in the other subunit (yellow in Fig. 5b) was only 0.30 ± 0.08 nm, which suggested that the bottom of E2 directly contacted the outer mouth of the selectivity filter and obstructed the extracellular ion conductive pathway of TREK-1 (Supplementary Fig. 8). In contrast, in the apo system $S_{apo}$ in which TKDC did not occupy the binding site, this distance was 0.56 ± 0.17 nm, indicating that helix E2 left a clear extracellular conductive pathway in the simulation (Fig. 5a and Supplementary Fig. 8).

**Other inhibitors targeting the identified pocket**. If the binding site of TKDC is reasonable, there may be more inhibitors recognizing this site. To further verify the binding site of TKDC in the extracellular cap, we performed a structure-based virtual screening targeting the identified site. More than 200,000

chemicals from the SPECS database were screened, and 25 hits were selected for bioassay (Fig. 6a). Two compounds with different chemotypes from TKDC (TKN1 and TKN2) displayed significant inhibitory effects on the current of TREK-1 (Fig. 6). Their $IC_{50}$ values were revealed to be 6.5 ± 1.7 μM and 3.8 ± 0.6 μM, respectively (Fig. 6c). These two compounds also showed inhibitory effects on TREK-2 and TRAAK (Supplementary Fig. 9 and Supplementary Table 3). These two chemotypes have not been reported as inhibitors of TREK channels. This result also demonstrates that the extracellular cap can be a binding target of inhibitors belonging to different chemotypes.

**Antidepressant-like effects of TKDC in mice**. To examine the antidepressant-like effect of TKDC, we chronically administered it to naive mice for 10 days. In the forced swimming test, the chronic administration of TKDC resulted in a decrease in immobility compared with the results in the vehicle-treatment group (Fig. 7a). In contrast, fluoxetine did not show antidepressant effects even at a higher dose (10 mg kg$^{-1}$). In the tail suspension test, we observed that TKDC induced a dose-dependent decrease in immobility (Fig. 7b). In the open field test, mice receiving TKDC were more likely to move into the central area (Fig. 7c). There was no obvious difference in the total distance traveled among groups, which excluded the possibility that the psychomotor substances may elicit false positive results (Fig. 7d). All tests showed that 10-day chronic treatment with TKDC is able to induce antidepressant-like effects at low doses (1 and 5 mg kg$^{-1}$), whereas fluoxetine treatment requires a longer time and higher doses. Similarly, a fast antidepressant-like effect of TKDC was also observed in the acute treatment (Supplementary Fig. 10).

**Discussion**

In this research, we show that TKDC is an inhibitor of TREK channels. Using this compound as a probe, we find that the extracellular caps of the TREK channels contain small-molecule binding sites. We apply a hybrid approach to validate the identified binding site from different perspectives. First, outside-out patch experiments indicate that TKDC acts directly on the extracellular side of TREK-1. Second, molecular docking is used to predict potential binding sites, and mutagenesis is applied to

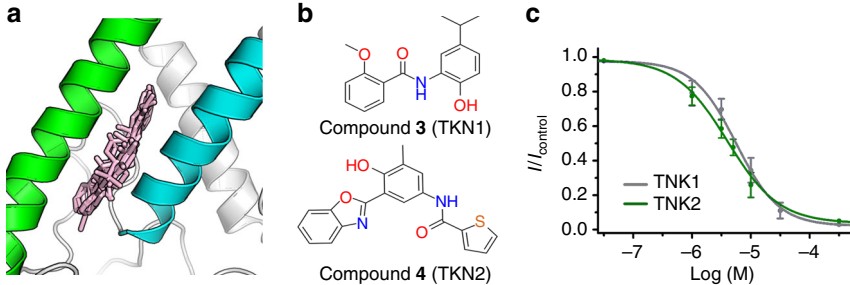

**Fig. 6** Inhibitors of TREK-1 identified in structure-based virtual screening. **a** Cartoon showing superimposed binding models of 25 hits. **b** Chemical structures of the discovered inhibitors of TREK-1. **c** Dose-dependent inhibition of TREK-1 by TKN1 and TKN2. $IC_{50}$ values were obtained via dose–response fitting

characterize the residues within the site. Third, a chemical probe derived from TKDC is designed to further investigate the extracellular binding site of the TRAAK channel and validate the proposed binding model. Finally, virtual screening is performed to discover more small-molecule active compounds targeting this site. The discovery of various inhibitors demonstrates that the extracellular cap may accommodate small-molecule modulators.

The extracellular ion pathway is functionally important in the ligand-induced inhibition of K2P channels. Earlier studies of K2P channels show that ruthenium red binds to the residues (D135 in TREK-2 and E70 in TASK-3) lining the wall of the extracellular ion pathway to inhibit TREK-2 and TASK-3[22–25]. Braun et al. reports that TREK-1 is not sensitive to ruthenium red. However, ruthenium red inhibits the I110D TREK-1 mutant, which has one isoleucine mutated to aspartate within its extracellular ion pathway[22]. In the current study, the proposed inhibitors, e.g., TKDC, do not interact directly with the residues within the extracellular ion pathway, but obstruct the pathway by inducing conformational changes of the extracellular helices. Our MD simulations suggest when TKDC binds to the groove consisting of the E1 and E2 helices, the E2 helix moves towards the selectivity filter and directly interacts with the pore region on the different subunit to block the extracellular ion pathway. An earlier crosslinking study of the other K2P channel TASK-3 by Clarke et al. shows that the E70 residue at the bottom of E2 helix and the H98 residue in the pore are close to each other[23]. By mutating one E70 residue and one H98 residue to cysteine in TASK-3, they found that crosslinking could occur between the Cys-70 and Cys-98 residues on different subunits of the dimer. This finding supports our assumption that the E2 helix can access the pore region and physically block the extracellular ion pathway in a ligand-induced allosteric transition.

In this work, we identify a TKDC-binding site in the extracellular cap. However, the proposed binding site is only shown in the crystal structure of TREK-1 (PDB code 4TWK), but not in the structures of the other K2P channels, e.g., TRAAK[13–15, 17]. Mutations of the residues A35, E38 and V42 within the binding site in the extracellular cap of TRAAK significantly increased the TKDC-induced inhibition, indicating that the proposed site was involved in modulating channels. Proteins are dynamic; crystal structures represent some of their numerous dynamic conformations. TKDC might insert into the extracellular cap of TRAAK in different conformations from the known crystal structures.

As an inhibitor targeting TREK-1, TKDC exhibited obvious antidepressant-like effects, which is consistent with previous studies showing antidepressant behaviors in TREK-1 knockout mice[34, 35]. In the animal behavior tests performed in this work, the antidepressant-like effects of TKDC were observed and compared with the effects of a known antidepressant drug, fluoxetine. Fluoxetine is well-known as an antagonist of the

5-HT$_{2B}$ receptor and can also inhibit TREK channels[7, 16]. The crystal structure of the TREK-2/fluoxetine complex suggests that fluoxetine binds to a fenestration adjacent to the pore filter entrance of TREK-2 (ref. [2]). Different from the binding site in the extracellular cap, this fenestration is highly conserved. The residues interacting with fluoxetine are mostly identical among TREK channels (Supplementary Fig. 11). In contrast, the residues composing the binding site in the cap are not conserved among the K2P family (Supplementary Fig. 11). Within the TREK subfamily, different members have varied residue compositions in the cap. Considering the unique residue E38 in TRAAK, we were able to design a chemical probe, 28NH, which lacks a negatively charged sulfonyl group and can significantly inhibit TRAAK ($IC_{50} = 11.8 \pm 2.1 \, \mu M$). Each K2P channel has distinctive features in the binding site of the extracellular cap. For example, TASK-5, THIK-1 and THIK-2 have multiple positively charged residues but few negatively charged ones in their binding sites. They might have high binding affinities to a ligand with a negatively charged group (Supplementary Fig. 11). TRAAK, TWIK-3 and TALK-2, which have more negatively charged residues in the binding sites, might favor a ligand with a positively charged group (Supplementary Fig. 11). The distinctive structural features of the proposed binding site can be directly applied to the design of specific inhibitors.

## Methods

**Molecular biology and cell transfection**. cDNA of the human TREK-1, TREK-2 and TRAAK channels were subcloned into the pEGFPN1 expression vector (Invitrogen), respectively. Mutations were introduced into TREK-1 or TRAAK by PCR using the QuickChange XL sitedirected mutagenesis kit (Agilent Technologies) and were subsequently confirmed by DNA sequencing.

Chinese hamster ovary (CHO) cells were provided by the Cell Bank of Chinese Academy of Sciences (Shanghai, China). CHO cells were maintained in DMEM/F12 (Gibco) supplemented with 10% fetal bovine serum (Gibco) and 100 μg ml$^{-1}$ penicillin-streptomycin (Cellgro) in a humidified incubator at 37 °C (5% CO$_2$). For electrophysiology experiments, cells plated in 35 mm tissue culture dishes (Corning Incorporated) were transiently transfected with the plasmids using PolyJet™ reagent (SignaGen) according to the manufacturer's instructions. Primers used in this study are listed in Supplementary Table 4.

**Electrophysiology**. All electrophysiological recordings were obtained using patch-clamp recordings in two different configurations: (1) whole cell for macroscopic current dose–response curves; and (2) outside-out and inside-out for macroscopic currents recordings.

All patch-clamp recordings were conducted 36–96 h after transfection with an Axopatch-200B amplifier (Molecular Devices) at room temperature (23–25 °C). The microelectrodes fashioned from 1.5 mm thin-walled borosilicate glass with filament were pulled from Flaming/Brown type micropipette puller (P-97; SUTTER INSTRUMENT) with the resistances of 3–7 MΩ (whole cell) or 7–15 MΩ (outside-out) when filled with a solution containing (in mM): 140 KCl, 2 MgCl$_2$, 10 EGTA, 1 CaCl$_2$, 10 HEPES (pH 7.3, adjusted with KOH). External solution contained (in mM): 150 NaCl, 5 KCl, 0.5 CaCl$_2$, 1.2 MgCl$_2$, 10 HEPES (pH 7.3, adjusted with NaOH). The current signals were filtered at 1 kHz and digitized at a 10 kHz sampling frequency by using DigiData 1440 A.

In the outside-out patch recordings and inside-out patch recordings, the inhibitor (10–30 μM TKDC) was applied via dish perfusion. The bath level was

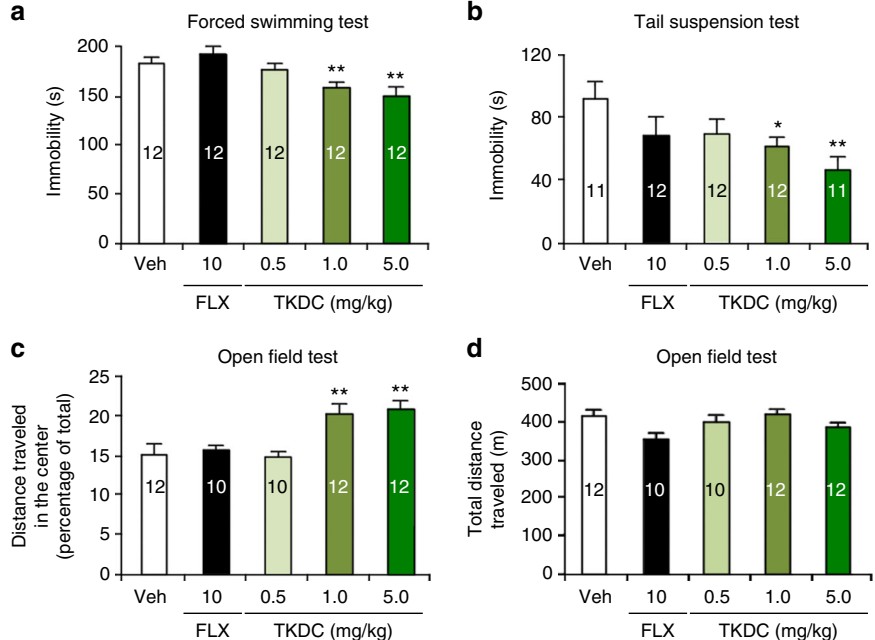

**Fig. 7** Chronic administration of TKDC in mice. **a** Time spent immobile in the forced swimming test after administration of TKDC and fluoxetine (FLX) [one-way ANOVA with post hoc LSD test, $F_{(4,55)} = 4.20$]. Veh indicates the vehicle-treatment group. TKDC was administered at doses of 0.5, 1 and 5 mg kg$^{-1}$. Fluoxetine was administered at a dose of 10 mg kg$^{-1}$. **b** Time spent immobile in the tail suspension test after administration of TKDC and fluoxetine [one-way ANOVA with post hoc LSD test, $F_{(4,53)} = 2.55$]. **c** Percentage of distance traveled in the center of the field over the total distance traveled after administration of TKDC and fluoxetine in the open field test (one-way ANOVA with post hoc LSD test, $F_{(4,51)} = 3.81$). **d** Total distance traveled after administration of TKDC and fluoxetine in the open field test (one-way ANOVA with post hoc LSD test, $F_{(4,51)} = 2.02$). The numbers in the bars indicate the number of cells studied per condition. The results are shown as the mean ± s.e.m; * indicates $P < 0.05$, ** indicates $P < 0.01$

kept as low as possible to reduce noise. Currents were recorded at a pipette potential of + 80 mV.

**Flow cytometry analysis**. CHO-K1 cells were transfected with pFLAG-TREK1 for 24 h. After washed with phosphate-buffered saline (PBS), the cells were disassociated with a non-enzyme cell dissociation solution (Sigma). After 10 µM TKDC incubation for 20, 40 min and 1 h at room temperature, cell surface TREK-1 expression in CHO-K1 cells was analyzed by flow cytometry. Transfected cells expressing FLAG-tagged TREK-1 were incubated with monoclonal ANTI-FLAG® M2-FITC antibody (Sigma). Data were collected with a flow cytometer (FACS Calibur, BD Bioscience) and analyzed with Flow Jo.

**Confocal microscopic analysis of channel internalization**. In eight-well chamber (Thermo), CHO-K1 cells were co-transfected with PEGFP-N1-TREK1 and OFP-Rab5a. After 24 h, cells were treated with 10 µM TKDC for 20 min at room temperature and DMSO as control. At the end of incubation, cells were washed with PBS, fixed with 4% paraformaldehyde and washed with PBS. Images were taken at ×100 with scan zoom 1.74 using Leica SP8 and 3D reconstruction was done using Imaris software (Bitplane) and co-localization analysis was quantified using Imaris software. An automatic threshold of red and green channels was selected. The data represent images taken from 18 different areas (n = 18).

**Chemicals**. All the tested compounds are known structures belong to our in-house library developed over time. The in-house library includes approximately 1000 small molecules purchased or synthesized for our previous and current research projects of drug discovery.

TKDC. (N-(4-cholorphenyl)-N-(2-(3,4-dihydrosioquinolin-2(1H)-yl)-2-oxoethyl)methanesulfonamide) and other compounds applied in virtual screening were purchased from SPECS. We synthesized 28NH, (2-((4-chlorophenyl)amino)-1-(3,4-dihydroisoquinolin-2(1H)-yl)ethanone).

To synthesize 28NH, we first synthesized 2-bromo-1-(3,4-dihydroisoquinolin-2(1H)-yl)ethanone. To a solution of 1,2,3,4-tetrahydroisoquinoline (980 mg) and N,N-dimethylaniline (932 mg, 1.1 equiv) in DCM (35 ml) was added 2-bromoacetyl bromide (1.49 g, 1.1 equiv), followed by stirring under ice-cooling for 30 min. After completion of the reaction, the solution was diluted with DCM (50 ml), water was added thereto, followed by extraction with DCM. The DCM layer was washed with brine and then dried over anhydrous sodium sulfate, and the solvent was removed by evaporation. The separation by silica gel chromatography column with petroleum ether/ethyl acetate (10/1) gave the desired product, compound 5 (1.42 g, 80%). ¹H NMR (400 MHz, CDCl₃) δ 7.21–7.19 (m, 4H), 4.70–4.51 (m, 2H),

4.23 (s, 2H), 3.72 (t, J = 5.8 Hz, 1H), 3.67 (t, J = 6.0 Hz, 1H), 2.91 (t, J = 5.7 Hz, 1H), 2.78 (t, J = 5.8 Hz, 1H).

Then, to a solution of 2-bromo-1-(3,4-dihydroisoquinolin-2(1H)-yl)ethanone (254 mg) and 4-chloroaniline (140 mg, 1.1 equiv) in DMF (5 ml) was added K₂CO₃ (276 mg, 2 equiv). The solution was heated for 1 h at 60 °C. The cooled solution was diluted with ethyl acetate (20 ml), water was added thereto, followed by extraction with ethyl acetate. The ethyl acetate layer was washed with brine and then dried over anhydrous sodium sulfate, and the solvent was removed by evaporation. The separation by silica gel chromatography column with petroleum ether/ethyl acetate (15/1) gave the desired product, compound 2 (147 mg, 49%). LC-MS: m/z (M + H) 300.9. ¹H NMR (400 MHz, CDCl₃) δ 7.33–7.12 (m, 6H), 6.58–6.55 (m, 2H), 4.98 (br, 1H), 4.78 (s, 1H), 4.60 (s, 1H), 3.93–3.87 (m, 3H), 3.66 (t, J = 5.9 Hz, 1H), 2.95 (t, J = 6.0 Hz, 1H), 2.89 (t, J = 6.0 Hz, 1H).¹³C NMR (101 MHz, CDCl₃) δ 167.79, 167.74, 146.02, 134.98, 133.89, 132.99, 131.74, 129.18, 129.08, 128.38, 127.33, 122.94, 126.92, 126.74, 126.68, 126.21, 122.19, 114.08, 45.99, 45.65, 45.55, 44.61, 42.09, 40.29, 29.28, 28.48; HPLC (Agilent HC-C18, CH₃CN: water = 70:30, flow rate = 1.0 mL min⁻¹, λ = 254 nm), purity = 97.2%.

**Molecular docking**. The crystal structures of TREK-1 (Protein Data Bank (PDB) code 4TWK), TREK-2 (PDB codes 4BW5, 4XDK, 4XDJ and 4XDL) and TRAAK (PDB code 4I9W, 4WFH, 4WFG, 4WFF, 4WFE, 4RUE and 4RUF) were used to detect ligand-binding pockets[1, 2, 4, 7]. TKDC was docked to each pocket in all structures using Schrodinger Glide software in SP mode with default parameters[28]. Pockets binding to TKDC with Glide G-scores below −3.0 were considered as possible binding sites. To validate the docking models generated using Glide, we also docked TKDC into all pockets using Autodock4 (ref. [29]). Searches were performed using Lamarckian Genetic Algorithm with default settings. Pockets binding to TKDC with Autodock binding scores below −6.0 were also considered as possible binding sites. Considering the energetic effects of the solvent environment and the receptor flexibility, RossettaLigand application[30–33] was applied to generate accurate molecular docking models of TKDC to the crystal structure of TREK-1 (PDB code 4TWK). TKDC was initially placed in the center of the proposed binding pocket identified in the Glide docking. Its center of mass was constrained to move within 1 nm diameter sphere, where it was allow moving freely during the docking process. Residue side chains in the binding site were repacked using a rotamer library. The docking model with the lowest binding energy was selected for analysis. Modeller[36] was used to create homology models of human wild-type TREK-2, wild-type TRAAK and three mutant TRAAK channels based on the crystal structure of TREK-1. In each case, model with the lowest root mean square deviation from the structure with PDB code 4TWK was applied to ligand docking using RossetaLigand application.

**Molecular dynamics simulations**. The crystal structure of TREK-1 (PDB code 4TWK) and the best docking model of TREK-1/TKDC were used to build the models of TREK-1 with and without TKDC. Missing atoms and loops were added back in most favored position without clashes using Discovery Studio 3.0[37]. The models of the TREK-1 with and without TKDC were inserted in a POPC (1-palmitoyl-2-oleoyl-sn-glycero-3-phosphocholine) lipid bilayer to establish a ligand-binding complex system and an apo system without TKDC, respectively. In each MD simulation system, the protein model and lipid bilayer were solvated in a periodic boundary condition box (10 nm × 10 nm × 13 nm) filled with TIP3P (transferable intermolecular potential 3 P) water molecules[38] and 0.15 M KCl. The apo system was replicated to generate two independent simulations. The ligand-binding system was replicated to generate three independent simulations. All MD simulations were performed using GROMACS version 4.52[39, 40], which employs the CHARMM all-atom force field[41, 42]. After 40 ns equilibration, a 1-μs-long production run was carried out for each apo system. For the ligand-binding system, a 40 ns equilibration and a 500 ns production were performed for each simulation. All productions were performed in the NPT ensemble at a temperature of 300 K and a pressure of 1 atm. Temperature and pressure were controlled using the velocity-rescale thermostat[43] and the Parrinello–Rahman barostat[44, 45] with isotropic coupling, respectively. Equations of motion were integrated with a 2 fs time step, and the LINCS algorithm was used to constrain bonds lengths[46]. Nonbonded pairlists were generated every 10 steps using a distance cutoff of 1.4 nm. A cutoff of 1.2 nm was used for Lennard-Jones (excluding scaled 1–4) interactions, which were smoothly switched off between 1 and 1.2 nm. Electrostatic interactions were computed using the Particle-Mesh-Ewald algorithm[47] with a real-space cutoff of 1.2 nm. Analysis of the simulations was performed using GROMACS and HOLE[48].

**Virtual screening for TREK-1**. A structure-based virtual screening strategy was performed using Schrodinger Glide software[28]. Approximately 200,000 commercially available compounds from the SPECS database were screened in the first run using the high-throughput virtual screening (HVS) module in Glide. The human TREK-1 crystal structure (PDB code 4TWK) was used as the docking receptor and residues in the extracellular cap are defined as the binding site. Glide G-scores was used to rank the result list and the top-ranked 20,000 candidates were selected. Since HVS applying simplified scoring function, these selected candidates were screened again using a standard docking module SP in Glide. The top-ranked 10,000 candidates were rescored by the CSCORE (consensus score) module of SYBYL 6.8 (Tripos Inc.). Compounds that had a consensus score of four or five or that were ranked in top 10% by at least four out of five scoring functions were selected. Using the Cluster Molecules module in Pipeline Pilot 7.5 (Scitegic, Inc.), these selected compounds were clustered based on their two-dimensional structures. To ensure the structural diversity of the compounds, two to three compounds with good drug-like properties (molecular weight < 500, log $P$ < 5 and polar surface area < 140 Å) were selected from each cluster for electrophysiological assay.

**Animal behavior tests**. Male C57BL/6 mice, aged 8 weeks (weighing 18–20 g) were used for acute and chronic drug administration and behavioral tests. The mice were housed four per cage under a 12-h light–dark cycle. Mice were raised under stable conditions with food and water ad libitum. All animal studies and experimental procedures were approved by the Animal Care Committees of the Shanghai Institute of Materia Medica, Chinese Academy of Sciences, and experiments were carried out in accordance with EU Directive 2010/63/EU on the protection of animals used for scientific purposes. For acute drug administration, mice were randomly divided into five groups and singly injected (i.p) with vehicle, 10 mg kg$^{-1}$ fluoxetine, 0.5 mg kg$^{-1}$ TKDC, 1 mg kg$^{-1}$ TKDC and 5 mg kg$^{-1}$ TKDC. For chronic drug administration, mice were also randomly divided into five groups and i.p. administrated with vehicle, 10 mg kg$^{-1}$ fluoxetine, 0.5 mg kg$^{-1}$ TKDC, 1 mg kg$^{-1}$ TKDC and 5 mg kg$^{-1}$ TKDC for 10 days (once per day). The mice were removed to a sound-attenuating testing room 3–4 h before testing. The behavioral tests were carried out 30 min and 24 h after acute and chronic drugs administration, respectively.

**Open field test**. The apparatus was made of opaque Plexiglas (50 cm × 50 cm) and placed in a room with sound proof. A camera was positioned above the apparatus to monitor the animal behaviors. The mice were randomly placed into the center of the sheet following 30 min drugs administration and their behaviors in the open field arena were recorded for 10 min using the camera. The surface of the sheet was cleaned with 75% ethyl alcohol after each trial to remove the permeated odors from previous animals. Locomotor activity and exploratory behavior were measured in the open field test and each animal was placed individually at the center for a 10 min session. The total distance moved and the number of rearing was measured in this experiment.

**Tail suspension test**. Mice were secured on the shelf by using adhesive tape placed approximately 1 cm from the tip of the tail. The mice were then suspended by placing the other end of the tape on the edge of shelf at the height of 80 cm above the floor. Six-minute session was performed for each animal. A blinded experiment was conducted to record the amount of time spent immobile during the last 4 min

of the testing period; the first 2 min were excluded. Because most mice are very active at the beginning of the test, and the potential effects of the treatment can be minimized during the first 2 min.

**Forced swimming test**. The mice were performed under normal light. The apparatus was a transparent Plexiglas cylinder with 30 cm height × 10 cm diameter and the water temperature at 23–25 °C were set to prevent mice from touching the bottom with their limbs. Mice were allowed to swim for 8 min, and their activity was videotaped. There was an extra 15-min swimming in the previous day. The duration of immobility defined as floating or remaining motionless was assessed. To avoid extremely active at the beginning of swimming, the first 2 min of swimming was also excluded as the tail suspension test.

**Drug administration**. For the in vitro study, the compounds (TKDC and 28NH) were dissolved in dimethyl sulfoxide. For the in vivo study, TKDC was dissolved in 5‰ tween-80 and intraperitoneally administered to animals. Used as a positive control, fluoxetine (Santa Cruz Biotechnology, USA) was dissolved in water intraperitoneally administered at a dose of 10 mg kg$^{-1}$. The vehicle, 5‰ tween-80, was administered to control mice as described above. All animals were treated with the drugs 30 min before the behavioral experiments.

**Data analysis and statistics**. Data of whole-cell voltage clamp experiments were fitted using doseResp function. The half inhibition concentrations were derived from fits of the dose–response curves to the function:

$$\frac{I}{I_0} = A_1 + \frac{A_2 - A_1}{1 + 10^{(\log x_0 - x)P}}, \tag{1}$$

where $I_0$ and $I$ are current amplitudes before and after application of inhibitors; $A_1$ and $A_2$ are constants between 0 and 1; $x$ is the concentration of inhibitor; $x_0$ is the concentration when 50% inhibition was reached (IC$_{50}$), $P$ is the Hill constant.

In statistical analyses, the experimental data were expressed as the mean ± s.e.m. The averaged IC$_{50}$ was collected from the fitting results of each patch (lasting for five inhibitor concentrations), and then was used to compare between different groups. Unless specified, an unpaired t-test was used when two means are compared; a one-way analysis of variance (ANOVA) along with post hoc LSD test was used when more than two means are compared. When the data distribution was skewed, the independent-sample Kruskal–Wallis test (non parametric test) followed by Dunn–Bonferroni post hoc test was performed. The results of animal behavioral tests (forced swimming test, tail suspension test and open field test) were analyzed using one-way ANOVA with post hoc LSD tests. $P < 0.05$ was considered statistically significant.

**Data availability**. The authors declare that the main data supporting the findings of this study are available within the article and its Supplementary Information files and available from the corresponding authors upon reasonable request.

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

## Acknowledgements

The research was supported in part by the National Natural Science Foundation of China (21422208), the Ministry of Science and Technology of the People's Republic of China (2013CB910601 and 2013CB910604), the "Personalized Medicines—Molecular Signature-based Drug Discovery and Development", Strategic Priority Research Program of the Chinese Academy of Sciences (XDA12040302), the National Key Scientific Instrument & Equipment Development Program of China (2012YQ0306010), the E-Institutes of Shanghai Municipal Education Commission (E09013) and the Special Program for Applied Research on Super Computation of the NSFC-Guangdong Joint Fund (the second phase) under Grant No.U1501501. We also thank the computer center of East China Normal University for computational resources.

## Author contributions

Q.L., L.C., X.C. and Y.M. conducted the majority of the experiments, including electrophysiological assays, molecular docking, molecular dynamics simulation and virtual screening. X.L. conducted the flow cytometry analysis and confocal microscopy analysis. B.Z., F.G. and L.L. carried out the animal behavioral experiments. S.Z. conducted the chemical synthesis experiments. L.C., X.C., Q.L., Y.M., Y.L. and H.Y. prepared the manuscript. Y.L. and H.Y. conceived and supervised the project. All authors approved the manuscript.

## Additional information

**Competing interests:** The authors declare no competing financial interests.

