## [Peer Review File · Nature Communications]

Reviewers' comments:

Reviewer #1 (Remarks to the Author):

Chen et al. use a variety of molecular tools to show that TREK has a binding site for inhibitor molecules at the extracellular cap of the channel protein. They have identified several inhibitors of TREK by screening and showed by mutagenesis that the molecules bind to a region between E1 and E2 region of the cap. They have identified specific amino acid residues that interact with the inhibitor. Furthermore, they have used TRAAK to support their finding that the cap contains a binding site at which inhibitor can reside and affect channel current.

Last year, Dong et al. found that fluoxetine binds to the region just below the selectivity filter by entering through the fenestration. This identified a region where an inhibitor such as norfluoxetine enter and cause conduction block. Chen et al now show that the cap region also has a binding site (at the region above the selectivity filter) that is accessible to small molecules that cause conduction block. Therefore, TREKs have two sites where inhibitors can bind and cause conduction block. An earlier study by Braun et al (BJP, 2015) showed that ruthenium red binds to the cap region to cause TREK block, indicating that a drug-binding site is present at the cap region. This study should be discussed and referenced, as it is the first demonstration of drug binding to the cap region.

This study by Chen et al. is a significant step forward for the understanding of the function of TREK and for identifying specific inhibitors, and should be of interest to others in the field. The inhibitors, however, would not be specific enough to distinguish between TREK1 and TREK2 whose tissue expression patterns vary greatly. The effects of these inhibitors on physiological function such as depression, pain and movements need to be carefully assessed to be sure which is caused by TREK1, TREK2 or both.

It is very surprising that the antidepressant effect was observed only 30 min after the administration of the inhibitor and fluoxetine. It is well known that many days of administration are usually necessary for antidepressant effect to occur with SSRI such as fluoxetine. This needs to be discussed clearly. What is the potential implication of this rapidly acting inhibitor and is this? A more careful and detailed study of the drugs on animal behavior after weeks of administration should be performed to understand if true antidepressant effect is observed. The Supplementary Figure 9 is too preliminary to be useful at this point, although positive data are shown.

There are a number of awkward sentences. One example is "...K2P channels are still lacking of the information...".

Many of the supplemental figures (i.e., 1, 4, 5, 6, etc) have contents that are similar to those of the figures in the manuscript itself, and may need to be revised or deleted. Also, In Fig. 1 and 4, the fitting of the curves (b and d) seems sufficient.

It is unclear how the data were fit. For example, in Fig. 1D, all data points are on the right side of the fitted curve (red), whereas all blue points are on the left side of the fitted curve. What equations were used to fit the data should be described, together with the precise statistical tests used, i.e., ANOVA with posthoc test, etc. for all figures.

Reviewer #2 (Remarks to the Author):

- Line 53 In these structures, K2P channels assemble as dimers – should be dimer of dimers. It is confusing how the authors describe it as it looks like there are 2 channels assembled rather than just one.
- Line 124 all available crystal structures of the TREK channels were applied to predict the binding sites of TKDC. English should be revised.
- Line 126 The GLIDE G-score presented the estimated free energy of binding, and seven binding site. Was the outcome the same using some other docking method?
- Line 128 In the sites 1, 2, 4, 5, 6 and 7, TKDC was surrounded by hydrophobic residues, which provided an unfavourable environment for the charged sulfonyl group of TKDC – choice of 3. Can site 3 be described with these other sites?
- Line 137 extracellular cap of TREK-1 was simulated using RossetaLigand application Why the other channels were not used and only TREK-1?
- Line 180 we built homology models of three TRAAK mutants (A35Q, E38T and V42Q) using the TREK-1 crystal structure as template. Why? Why the crystal structures of TRAAK were not used? There are plenty to choose: TRAAK (PDB code 4I9W, 4WFH, 4WFG, 4WFF, 4WFE, 407 4RUE and 4RUF) What is the sequence homology between TREK-1 and TRAAK? Are these homology models justified?
- Investigation of ligand-binding mode of TRAAK using a chemical probe. English needs to be checked.
- Line 224 28NH was tolerated by E38 and inserted deeply into the extracellular cap of TRAAK What do the authors mean by '28NH was tolerated by E38'?
- Line 240 Three independent molecular dynamics (MD) simulations were performed for the complex system Scomplex for 500 ns, respectively. To investigate the influence of TKDC, an apo system Sapo without TKDC was established as a control under the same condition as Scomplex, and applied to perform a 1- μ s long simulation English needs revision.
- Line 243 TKDC interacted with I80, L102 and I106 at a high frequency (> 0.85). What is the meaning of >0.85?
- Line 247 During the last 200-ns simulations, distinct conformations were observed in the apo system Sapo and the complex system Scomplex (Fig. 5). Conformations referred to the ligand or the channel? It is not clear.
- Line 276 This result also demonstrates that the extracellular cap may accommodate various inhibitors. What does this mean? At the same time? At the same location?
- Line 293 Third, a chemical probe derived from TKDC was used to validate the proposed binding mode. Why TKDC was not used?
- Line 298 In this work, we identified the TKDC-binding site in the extracellular cap. However, the proposed binding site is only shown in the crystal structure of TREK-1 (Protein Data Bank (PDB) code 4TWK). Our mutagenesis study of TRAAK ruled out this possibility. Substitutions of the residues A35, E38 and V42 within the binding site in the extracellular cap of TRAAK significantly increased the TKDC-induced.

If the existence of a binding site in TREK-1 was already reported in the literature, what is the novelty of this study?
- Line 411 Ligand dockings of TKDC to TREK-1 were performed using both GLIDE software³¹ and RossetaLigand application. Why this was done only on TREK-1 and not the rest of

channels considered? It is as if the studies are half-done. For some things one channel is used, for some others, the authors use other channels. It is fine if the choices are justified, but they are not.

- Line 414 Modeller was used to create homology models of human wildtype (WT) and three mutant TRAAK channels based on the crystal structure of TREK-1. Already I asked earlier about the reason behind this choice.
- Line 427 filled with TIP3P - should be filled.
- Abstract: [...] And molecular dynamics simulations revealed the ligand-induced allosteric conformational transitions which obstructed the ion conductive path. Can this really be concluded from just one simulation?
- Line 25 diseases. However, K2P channels are still lacking of the information of druggable site. **However, information of the druggable sites in K2P channels is still lacking.**
- Line 29 an allosteric ligand-binding site in the extracellular cap of the channels. **And** molecular dynamics
- Line 31 the ion conductive path. The presence of identified ligand-binding site in cap was confirmed by **The identification of the ligand-binding site in the cap was confirmed by**
- Line 33 that **the** extracellular caps of K2P channels can act as **a** new allosteric site and serve as direct drug
- Line 46 and **plays roles is involved** in glutamate conductance and the regulation of blood-brain barrier
- Line 64 contribute to distinct traits in the way K2P channels interact with ligands. For example, **the cap**
- **Line 65 prevents direct ion-transit between the pore mouth and the extracellular milieu, which explains the refractoriness of K2P channels to inorganic and toxin pore blockers 22,23.** It is not very clear on what refractoriness means in this context. Maybe insensitivity? If so, I would change this word add something at the end like 'that traditionally block K⁺ channels'
- Line 68 against other ion channels. **However, how the cap responds to ligands is still unclear.**
- Line 77 the extracellular cap of K2P channels is **a** functionally important **as a** drug targets.
- Line 136 To accurately dock TKDC **to site 3**, considering the receptor flexibility and environmental effects, the
Just to make it clear that this was the only binding site to be analysed further from the blind dock, if this interpretation is correct.
- Line 137 extracellular cap of TREK-1 was simulated using **RosettaLigand application24-27.** Is it necessary to say this here? This is also mentioned at least 3 times in the text.
- Line 159 (a) Binding site of TKDC in the extracellular cap of TREK-1 channel. (b) Detailed **show view** of
- Line 286 K2P channels represent important potential drug targets for various clinical treatments. Identifying ligand-binding sites in K2P channels is therefore of great importance. The utilization of chemical probes could facilitate the characterization of ligand-binding sites.
This is really vague and said in both abstract and intro already; is it necessary?
- Line 298 In this work, we identified the TKDC-binding site in the extracellular cap. However, the proposed binding site is only shown in the crystal structure of TREK-1 (Protein Data Bank

(PDB) 300 code 4TWK), not in those of the other K2P channels, e.g., TRAAK14,15,17,18. Does TKDC modulate TRAAK by the other domains rather than the extracellular cap?

- Line 302 of TRAAK ruled out this possibility. Substitutions of the residues A35, E38 and V42 within the binding site in the extracellular cap of TRAAK significantly increased the TKDC-induced inhibition, indicating the proposed site is involved in modulating channels. Proteins are dynamic, and crystal structures represent some of their numerous dynamic conformations. The TKDC might insert into the extracellular cap of TRAAK in conformations different from the known crystal structures.

The mutagenesis data confirms that this site is important, but it does not really rule out that the ligand can bind elsewhere and have minor effects from other sites in TRAAK, as this has not been explored at all. Also, it is described very clearly the residual differences that cause the reduced sensitivity of TRAAK (and why) by both mutagenesis and docking in the results so it seems contradictory to just say now that the protein is dynamic and the binding site may present itself in other conformations of the protein. To me, it makes more sense to align the sequences of more K2P channels to see if the important residues are conserved to predict if any other channels may be selective to this ligand.

- Line 309 As an inhibitor targeting TREK-1, TKDC exhibited obvious antidepressant-like effects (supplementary Fig. 9), which is consistent with previous behavioral tests indicating an antidepressant effect when knocking out the *Trek1* gene in mice^{28,29}

Maybe I do not understand the experimental results, but these experiments do not seem to be mentioned in the results.

- Line 312 behavior tests performed in this work, the antidepressant-like effects of TKDC were observed and compared to a known antidepressant drug, fluoxetine. Well-known as an antagonist of the 5-HT_{2B} receptor, fluoxetine can also inhibit TREK channels^{6,13}.

I understand that they authors have compared it to fluoxetine because it is a known inhibitor, but it does not seem to add any extra insight as they make it very clear it is very different in terms of location and sequence. I would prefer a discussion of how this site apparently physically blocks the channels via the cap, as this seems more relevant to me and unique to K2P channels.

Reviewer #3 (Remarks to the Author):

This is an interesting study that uses a combination of electrophysiological profiling, chemistry, mutagenesis, molecular modeling and virtual screening to identify novel classes of potentially selective (and in vivo active) modulators of the TREK subfamily of K2P channels. Overall, the findings are interesting and the combination of approaches largely complementary in supporting the authors' main conclusions.

1. The initial description of the "fortuitous" identification of TKDC from an internal library of ~1000 molecules needs a little more elaboration. What library are the authors referring to? Presumably an in-house library that has been developed over time, or was it curated in a particular manner etc?

2. Determination of inhibitor potencies and statistical comparisons. How was the potency for TKDC established at TRAAK, given that it had very small effects up to the highest concentrations tested? (Fig. 1d). The authors have clearly assumed a complete inhibition to be able to derive an IC50 value, but this should be explicitly stated as a caveat. More importantly, all potency estimates are given as IC50- +/- SEM, but it is well established that drug potency (or affinity) values are only Gaussian (or approximately so) when expressed as logarithms. Thus, ALL the statistics shown in the manuscript are actually invalid (t tests and ANOVAS assume a Gaussian distribution), unless they have actually been performed on the logarithms of the potency values. This needs to be clarified and corrected.

3. In the first description of the modeling results in the main manuscript, when discussing the proposed binding mode of TKDC to TREK-1, a little more elaboration is required for the reader to be able to better understand. Rather than simply stating "Using the molecular docking method...", The authors need to be more explicit. Based on my reading of the Methods, the authors actually adopted a two-stage approach, is that correct? In the first stage, Schrodinger GLIDE using default parameters was applied to the TREK family crystal structures to identify a variety of possible poses across the entire family. Based purely on the docking scores from these studies (< - 8.0), the seven highest ranked poses were chosen. The second stage of the modeling involved both GLIDE and RosettaLigand to more accurately understand the proposed docking of TKDC into TREK-1 (in what was essentially "Site 3" from the first stage of modeling). This two-stage approach should be explained better in the Results.

4. While still on the modeling, I find it quite interesting that 5 of the 7 highest ranked poses were from the TRAAK structures, despite the fact that TKDC had lowest potency for this receptor! The authors make a reasonable case for ruling out the poses and settling on Site 3 but what would happen, for instance, if they looked at Site 9, which was also in TREK-1, docked at a lower site than Site 3 and with a slightly lower score? I guess my main concern here is the rather arbitrary nature in using initial docking scores (especially since most of the best scores come from the "wrong" channel) to choose the right one. Of course, much of this concern is mitigated by many of the subsequent experiments performed by the authors, but it is surprising that the two receptors for which TKDC shows highest potency, TREK-1 and TREK-2, yielded that fewest high ranked poses. I discuss some questions regarding TREK-2 below.

5. The mutational validation of the EC cap in TREK-1 seems convincingly done, but I am surprised as to why the authors did not mutate T79. Mutation of this residue should be tested, given that all the other residues were tested (including Q83 and Q105, which had no effect upon alanine substitution).

6. The introduction of key TREK-1 residues into TRAAK (Figure 3) provides good support for the role of the extracellular cap in inhibitor action. However, the resulting curves appear distinctly biphasic to me. Does this mean that the modulator can actually bind to more than one site? This is of relevance due to one of the points the authors themselves make in the Discussion.

7. With regards to the MD simulations (Fig. 5), are there any existing data from other studies of this channel class (e.g., mutagenesis or cross-linking, for instance) to support the possibility of an allosteric transition of E2 similar to that being proposed to be caused by the binding of TKDC? Also, the authors are highlighting the final pose at the end of the MD run, but what does the starting pose look like? They discuss interaction frequencies, but it may also help visually to at least see how the compound started relative to how it ends to further highlight stability.

8. The use of a virtual screen to identify additional classes of inhibitor is an additional strength of this MS. However, I am struck by the fact that the authors did not do a fairly obvious experiment, which is to test them (at least) on TREK-2 and TRAAK. This also speaks to the issue of the potential for selectivity vs off-target activity. I accept that the latter may be asking too much, but the acknowledgment of the potential for other target effects should be addressed, at least in future studies. However, I do feel that selectivity testing across TREK-2 and TRAAK is required.

9. Finally, I may have missed this, but I am surprised to see a lack of docking of any of the inhibitors, at least TKDC, into TREK-2. Based on the original assay, this had as good, if not better, potency at TREK-2 than TREK-1, yet the latter is used as the template for all subsequent studies due to the identification of "Site 3". At the very least the authors need to be able to explain the effect on TREK-2 in terms of their proposed model, or else need to suggest an alternative site?

Reviewer #4 (Remarks to the Author):

K2P family channels control cellular electrical excitability in a wide range of physiological contexts. They generate background K⁺ currents to set and maintain the resting membrane potential and are additionally regulated by a diverse set of modulators. Pharmacological modulation of their activity would provide a means to dissect the biological roles of K2Ps and has potential for clinical applications in numerous contexts. However, development of specific pharmacology has been difficult and remains an important goal. In this manuscript, Chen, Cheng, Luo et. al identify a new small-molecule blocker of TREK subtype K2Ps called TKDC. They use a combination of electrophysiological and computational approaches to present a model for block by TKDC through binding to a unique extracellular cap domain in the channel. In addition, they provide evidence from behavioral studies consistent with its action on TREK channels in vivo. This work is overall nicely done and very interesting. I offer below some suggestions for consideration that I believe would improve the manuscript and make the findings immediately more useful for the field:

1. Add an experiment testing block of channel by TKDC of TREK-1 or -2 in inside-out vs. outside-out patches. This would provide additional evidence that block is truly direct and could demonstrate that inhibition strictly requires access to the extracellular cap domain as predicted by the model presented.

2. Does drug binding effect activation of the channels in addition to basal activity (i.e. by mechanical force, lipids, or pH change)? A comprehensive survey would be nice, but even evaluating block +/- one activating stimulus would be a very useful piece of information to know for those interested in using this drug.

3. Does TKDC block other K2Ps? Testing block of at least one K2P from each clade of the family would accomplish two things: it would be a complementary test of the mechanism proposed and it would make the drug immediately more useful to the field if its specificity among the entire K2P family is characterized.

4. Both binding and washout of TKDC appear slow in electrophysiological recordings (Fig. S1D). A discussion of why this is would be interesting. Is it consistent with the molecular dynamics simulations/crystal structures and indicative of an infrequently populated conformation that exposes the binding site to solution for drug access?

5. The presentation of the antidepressant-like effects of TKDC in vivo should be moved out of the discussion and into the main text.

The statistical analysis used are appropriate.

Responses to the comments of reviewers

Reviewer #1:

Chen et al. use a variety of molecular tools to show that TREK has a binding site for inhibitor molecules at the extracellular cap of the channel protein. They have identified several inhibitors of TREK by screening and showed by mutagenesis that the molecules bind to a region between E1 and E2 region of the cap. They have identified specific amino acid residues that interact with the inhibitor. Furthermore, they have used TRAAK to support their finding that the cap contains a binding site at which inhibitor can reside and affect channel current.

Comment 1: Last year, Dong et al. found that fluoxetine binds to the region just below the selectivity filter by entering through the fenestration. This identified a region where an inhibitor such as norfluoxetine enter and cause conduction block. Chen et al now show that the cap region also has a binding site (at the region above the selectivity filter) that is accessible to small molecules that cause conduction block. Therefore, TREKs have two sites where inhibitors can bind and cause conduction block. An earlier study by Braun et al (BJP, 2015) showed that ruthenium red binds to the cap region to cause TREK block, indicating that a drug-binding site is present at the cap region. This study should be discussed and referenced, as it is the first demonstration of drug binding to the cap region.

Response: Thank the reviewer for the valuable advice. Braun *et al.* reports that ruthenium red binds to the residues (D135 in TREK-2 and E70 in TASK-3) lining in the wall of the extracellular ion pathway to inhibit TREK-2 and TASK-3. In our work, the proposed inhibitors of TREK channels, e.g. TKDC, do not directly interact with the residues within the ion pathway, but obstruct the pathway by inducing conformational changes of the extracellular helices. We have revised the manuscript to add more discussion about the earlier studies of ruthenium red and have referenced them accordingly.

Comment 2: This study by Chen et al. is a significant step forward for the understanding of the function of TREK and for identifying specific inhibitors, and should be of interest to others in the field. The inhibitors, however, would not be specific enough to distinguish between TREK1 and TREK2 whose tissue expression patterns vary greatly. The effects of these inhibitors on physiological function such as depression, pain and movements need to be carefully assessed to be sure which is caused by TREK1, TREK2 or both.

Response: Genetic ablation of TREK-1 correlates with many neurophysiological and neurobehavioral phenotype, including ischemia, epilepsy, pain and depression. In this work, we pay attention to the important functional role of TREK-1 in the field of depression and illustrate that TKDC has antidepressant-like properties on behavioral level. Previous animal behavioral tests show that the deletion of TREK-1 results in depression-resistant phenotypes in mice [1]. In addition, a natural peptide spadin specifically blocks the TREK-1 and displays antidepressant properties [2]. These studies provide compelling evidence suggesting TREK-1 as a target for antidepressants. In contrast, a recent neurobehavioral study of TREK-2 shows no genotype-dependent differences between the wild type and TREK-2-knockout mice in the open field test, which suggests TREK-2 may be not involved in the

anti-depressant responses [3].

The main aim of this work is to reveal an allosteric ligand-binding site in the extracellular cap of K2P channels. We acknowledge that TKDC is not specific enough to distinguish between TREK-1 and TREK-2. But the identified extracellular binding site provides structural information to discover and design more specific inhibitors for TREK-1 or TREK-2 in the future. Those highly specific inhibitors can be better pharmacological tools in the assessment of physiological functions of TREK-1 and/or TREK-2.

Reference:

[1] Heurteaux C, *et al.* (2006) Deletion of the background potassium channel TREK-1 results in a depression-resistant phenotype. *Nature neuroscience* 9(9):1134-1141.

[2] Borsotto M, *et al.* (2015) Targeting two-pore domain K(+) channels TREK-1 and TASK-3 for the treatment of depression: a new therapeutic concept. *British journal of pharmacology* 172(3):771-784.

[3] Mirkovic K, Palmersheim J, Lesage F, & Wickman K (2012) Behavioral characterization of mice lacking Trek channels. *Frontiers in behavioral neuroscience* 6:60.

Comment 3: It is very surprising that the antidepressant effect was observed only 30 min after the administration of the inhibitor and fluoxetine. It is well known that many days of administration are usually necessary for antidepressant effect to occur with SSRI such as fluoxetine. This needs to be discussed clearly. What is the potential implication of this rapidly acting inhibitor and is this? A more careful and detailed study of the drugs on animal behavior after weeks of administration should be performed to understand if true antidepressant effect is observed. The Supplementary Figure 9 is too preliminary to be useful at this point, although positive data are shown.

Response: Regarding the acute administration of TKDC, since the drug brain level is abundant 30 minutes (on target) after single i.p. injection (e.g., 31290 ng/g for fluoxetine in the brain), 30-60 minutes is an appropriate time window to preliminarily screen compounds' antidepressant effect (including SSRIs) [1]. In the acute treatment, our data suggest the potential antidepressant effect of TKDC (supplementary Fig. 10). 24 hours after drug administration, the brain drug level would have dramatically decreased (e.g., 6302 ng/g for fluoxetine in the brain) and the antidepressant effect of most SSRIs will be eliminated. A long-term (24-hour) antidepressant efficacy of SSRIs usually requires the chronic administration for more than 14 days.

In this revision, to examine the antidepressant-like effect of TKDC, we chronically administered it to naïve mice for 10 days. In the forced swimming test, the chronic administration of TKDC resulted in a decrease in immobility, compared to the vehicle-treatment group (Fig. 7a). In contrast, fluoxetine did not show antidepressant effects even at a higher dose (10 mg/kg). In the tail suspension test, we observed TKDC induced a dose-dependent decrease in immobility (Fig. 7b). In the open field test, mice were more likely to move into the central area in the TKDC administration (Fig. 7c). There was no obvious difference of the total distance traveled among groups, which excluded the possibility that the psychomotor substances may elicit false positive results (Fig. 7d). All of these tests indicate that 10-day chronic treatment with TKDC is able to induce antidepressant-like effects at low doses (1 mg/kg and 5mg/kg), while fluoxetine treatment requires a longer time and higher doses. We have revised the manuscript to include this information in the results section.

Supplementary Figure 10 | Antidepressant characterization of TKDC

(a) The statistical analysis of total distances traveled after drugs administration reflection of the locomotion activity in the open field test [one-way ANOVA with *post-hoc* LSD test, $F(4,50) = 1.58$]. Veh represented the vehicle. FLX represented the fluoxetine administered at a dose of 10 mg/kg. TKDC was administered at a dose of 0.5 mg/kg, 1 mg/kg and 5 mg/kg, respectively. (b) Histogram indicated the number of rearing behaviors in the open field test [one-way ANOVA with *post-hoc* LSD test, $F(4,53) = 3.15$]. (c) Histogram indicated the duration of immobility in the forced swimming test [one-way ANOVA with *post-hoc* LSD test, $F(4,53) = 8.90$]. (d) Histogram indicated the duration of immobility in the tail suspension test [one-way ANOVA with *post-hoc* LSD test, $F(4,49) = 2.71$]. The numbers in the bars indicate the number of cells studied per each condition. Results are shown as mean \pm SEM; * indicates $P < 0.05$, ** indicates $P < 0.01$.

Figure 7. Chronic administration of TKDC in mice.

(a) Time spent immobile in the forced swimming test after administration of TKDC and fluoxetine (FLX) [one-way ANOVA with *post-hoc* LSD test, $F(4,55) = 4.20$]. Veh indicates the vehicle-treatment group. TKDC was administered at doses of 0.5 mg/kg, 1 mg/kg and 5 mg/kg, respectively. Fluoxetine

was administered at a dose of 10 mg/kg. (b) Time spent immobile in the tail suspension test after administration of TKDC and fluoxetine [one-way ANOVA with *post-hoc* LSD test, $F(4,53) = 2.55$]. (c) Percent distance traveled in the center of the field over the total distance traveled after administration of TKDC and fluoxetine in the open field test [one-way ANOVA with *post-hoc* LSD test, $F(4,51) = 3.81$]. (d) Total distance traveled after administration of TKDC and fluoxetine in the open field test [one-way ANOVA with *post-hoc* LSD test, $F(4,51) = 2.02$]. The numbers in the bars indicate the number of cells studied per each condition. Results are shown as mean \pm SEM; * indicates $P < 0.05$, ** indicates $P < 0.01$.

Reference:

[1] Holladay JW, Dewey MJ, & Yoo SD (1998) Pharmacokinetics and antidepressant activity of fluoxetine in transgenic mice with elevated serum alpha-1-acid glycoprotein levels. *Drug metabolism and disposition: the biological fate of chemicals* 26(1):20-24.

Comment 4: There are a number of awkward sentences. One example is "...K2P channels are still lacking of the information...".

Response: We are sorry for the grammatical errors in our manuscript. We have polished the manuscript with a professional assistant in writing.

Comment 5: Many of the supplemental figures (i.e., 1, 4, 5, 6, etc) have contents that are similar to those of the figures in the manuscript itself, and may need to be revised or deleted.

Response: Thank the reviewer for the valuable advice. We have revised the figures and deleted the redundant ones.

Comment 6: Also, In Fig. 1 and 4, the fitting of the curves (b and d) seems sufficient. It is unclear how the data were fit. For example, in Fig. 1D, all data points are on the right side of the fitted curve (red), whereas all blue points are on the left side of the fitted curve. What equations were used to fit the data should be described, together with the precise statistical tests used, i.e., ANOVA with posthoc test, etc. for all figures.

Response: We appreciate this suggestion. We have performed more independent whole-cell voltage clamp experiments, and have fitted all previous and new data using doseResp function in OriginPro 8.1 software (Fig. 1d and Fig. 4b). The half inhibition concentrations were derived from fits of the dose-response curves to the function:

$$\frac{I}{I_0} = A_1 + \frac{A_2 - A_1}{1 + 10^{(\log x_0 - x)P}}$$

Where I_0 and I are current amplitudes before and after application of inhibitors; A_1 and A_2 are constants between 0 and 1; x is the concentration of inhibitor; x_0 is the concentration when 50% inhibition was reached (IC_{50}), P is the Hill constant.

In statistical analyses, the experimental data were expressed as mean \pm SEM. Origin 8.1 software (OriginLab Corporation, Northampton, USA) was used in the analyses of the cellular experiments. The averaged IC_{50} was collected from the fitting results of each patch (lasting for five inhibitor concentrations), and then was used to compare between different groups. Unless specified, an unpaired t test was used to compare two means and a one-way ANOVA along with post-hoc LSD test was used to compare three or more mean. When the data distribution was skewed, the independent-sample Kruskal-Wallis test (non parametric test) followed by Dunn-Bonferroni post hoc test was performed. The results of animal behavioral tests (forced swimming test, tail suspension test and open field test) were analyzed using one-way ANOVA with post-hoc LSD tests in SPSS software. $P < 0.05$ was considered statistically significant. We have revised the manuscript to include this information for all figures, and have added more descriptions in the Methods.

Figure 1. Inhibition of TREK subfamily channels by TKDC in CHO cells.

(a) Chemical structure of TKDC. (b, c) Typical whole-cell current traces recorded from CHO cells overexpressing the TREK-1 channel with 10 μ M TKDC (b) or DMSO application (c). Currents were elicited by depolarizing voltage steps from a holding potential of -80 mV to +80 mV in 20 mV increments and then stepping down to -60 mV. (d) Dose-dependent inhibition of TKDC on TREK-1, TREK-2 and TRAAK channels. (e) The statistics of the half-inhibitory concentrations of TKDC to TREK-1 ($n = 7$), TREK-2 ($n = 7$), and TRAAK ($n = 8$) channels. IC_{50} values were obtained via dose-response fitting. Kruskal-wallis test was used for statistical analysis; ** indicates $P < 0.01$. The data are shown as mean \pm SEM.

Figure 4. Exclusion of the sulfonyl group of TKDC enhanced inhibitory effect on TRAAK.

(a) Chemical structure of 28NH. (b) Dose-dependent inhibition of TRAAK by TKDC and 28NH. (c) The statistics of IC_{50} values for the inhibition of TREK-1, TREK-2, TRAAK channels by TKDC and 28NH. IC_{50} values were obtained via dose-response fitting. Unpaired t-test was used for statistical analysis [$t(7) = 1.027$ and $n = 9$ for TREK-1, $t(4) = 0.910$ and $n = 6$ for TREK-2, and $t(6) = 5.724$ and $n = 8$ for TRAAK]; ** indicates $P < 0.01$. The numbers in the bars indicate the number of cells studied per each condition. The data are shown as mean \pm SEM. (d) Binding model of 28NH to TRAAK. The protein is shown as a cartoon. 28NH and key residues are shown as sticks.

Reviewer #2 (Remarks to the Author):

Comment 1: Line 53 In these structures, K2P channels assemble as dimers – should be dimer of dimers. It is confusing how the authors describe it as it looks like there are 2 channels assembled rather than just one.

Line 124 all available crystal structures of the TREK channels were applied to predict the binding sites of TKDC. English should be revised.

Response: We are sorry for the grammatical errors in our manuscript. We have polished the revised manuscript with a professional assistant in writing.

Comment 2: Line 126 The GLIDE G-score presented the estimated free energy of binding, and seven binding site. Was the outcome the same using some other docking method?

Response: Yes. We also docked TKDC to the TREK family crystal structures using Autodock. The outcome generated using Autodock was highly consistent with the results generated using Glide. Compared with the other sites, TKDC binds to the proposed TKDC binding site (site 3) with the lowest Autodock binding score. Particularly, the Autodock docking model of TKDC in the site 3 was very similar to the Glide docking model. We have added this data as Figure 3 and Figure 4 in the supplementary, and have revised the section of “binding mode of TKDC to TREK-1” accordingly.

Supplementary Figure 3 | Docking of TKDC to TREK channels

(a) Potential ligand-binding sites in the crystal structures of TREK channels. Each predicated binding site is indicated as a sphere. (b) Glide G-scores and Autodock binding scores of docking TKDC to each potential binding site. The binding sites were assigned in the different groups according to their locations and compositions of residues. Eleven different groups are shown in different colors.

Supplementary Figure 4 | Docking models of TKDC in the binding sites of groups A and B generated using Glide and Autodock

(a-f) Representative docking poses of TKDC in the sites of group A, including (a, b) sites 1, 4, 5, 6, 7 and 8, (c, d) sites 11 and 21, and (e, f) site 2. (g, h) Representative docking poses of TKDC in the site 3 of group B. These models were generated using (a, c, e, g) Glide and (b, d, f, h) Autodock. TKDC and the protein residues in the binding sites are shown as sticks. The hydrophobic residues interacting with the charged sulfonyl group of TKDC are highlight in yellow. Residues blocking view are omitted.

Comment 3: Line 128 In the sites 1, 2, 4, 5, 6 and 7, TKDC was surrounded by hydrophobic residues, which provided an unfavorable environment for the charged sulfonyl group of TKDC – choice of 3. Can site 3 be described with these other sites?

Response: Thank the reviewer to point out that the binding sites predicted by the docking methods were not described clearly. In fact, some of the 36 predicted binding sites were equivalent sites in the different crystal structures. For example, the sites 1, 2, 4, 5, 6, 7, 8, 11 and 21 were all binding pockets

consisting of M1, M2, M4 and P1 helices and shared conservative residues. According to their locations and compositions of residues, now we assigned all predicated binding sites into eleven groups (group A-K). All sites were ranked based on the Glide G-score and Autodock binding score, respectively. Regardless of docking methods (Glide and Autodock), the predicated binding sites in groups A and B always had the lowest scores (Glide G-scores < -7.5 or Autodock binding score < -9.0) (supplementary Fig. 3b in the **Response to Comment 3**). In the sites of group A (sites 1, 2, 4, 5, 6, 7, 8, 11 and 21), TKDC was surrounded by hydrophobic residues, which provided an unfavorable environment for the charged sulfonyl group of TKDC (supplementary Fig. 4a-f in the **Response to Comment 3**). Therefore, the sites of group A may not bind to TKDC. In the group B, there was only one member, i.e., site 3, which had the third lowest G-score and the lowest Autodock binding score. Site 3 was located in the extracellular cap domain consisting of two helices from different subunits in TREK-1. As an extracellular groove observed only in a crystal structure of TREK-1 (Protein Data Bank (PDB) code 4TWK), it has not been reported to bind to the inhibitors of K2P channels. In the site 3, TKDC showed a more reasonable pose, in which its hydrophobic groups were inserted into a gap between extracellular helices and the polar sulfonyl group was exposed to the extracellular solvent (supplementary Fig. 4g-h in the **Response to Comment 3**). Thus, site 3 was considered as the possible binding site of TKDC in TREK-1, and was selected for further analysis. We have rewritten this part in the section of “binding mode of TKDC to TREK-1”.

Comment 4: Line 137 extracellular cap of TREK-1 was simulated using RossettaLigand application. Why the other channels were not used and only TREK-1?

Response: TREK-1, TREK-2 and TRAAK belong to the same subfamily and TKDC can inhibit all of them. We studied the ligand binding of TREK-1 and TRAAK, in which TRAAK showed a much lower sensitivity to TKDC than TREK-1 did. TREK-2 and TREK-1 are highly homologous and TKDC shows a good potency at both of them. Therefore, in the initial study, we only selected TREK-1 as a representative channel to study the ligand-binding site.

In the revision, we have docked TKDC to the proposed extracellular binding site of TREK2 using RossettaLigand. The TREK-2/TKDC docking model with the lowest binding energy (supplementary Fig. 5) was very similar to the TREK-1/TKDC docking model. In the binding site of TREK-2, the bicyclic group of TKDC was fully enveloped by a cavity formed by residues I105, Q108, L127 and H130. Close to the bottom of the extracellular cap domain, the chlorophenyl group of TKDC was inserted in a cavity formed by Q101, T104 and I105. The negatively charge sulfonyl group of TKDC was attracted by the adjacent positively charged H130, which might stabilize the binding of TKDC to this site.

Supplementary Figure 5 | Binding mode of TKDC to TREK-2

TKDC and the residues in the extracellular binding site are shown as sticks. The E1 and E2 helices are shown as green and cyan cartoons.

Comment 5: Line 180 we built homology models of three TRAAK mutants (A35Q, E38T and V42Q) using the TREK-1 crystal structure as template. Why? Why the crystal structures of TRAAK were not used? There are plenty to choose: TRAAK (PDB code 4I9W, 4WFH, 4WFG, 4WFF, 4WFE, 4RUE and 4RUF). What is the sequence homology between TREK-1 and TRAAK? Are these homology models justified?

Response: Our mutagenesis study of TRAAK suggests three residues (A35, E38 and V42) contribute to binding TKDC to the proposed binding site. However, this proposed site is only shown in a crystal structure of TREK-1 (PDB code 4TWK), but is absent in the available crystal structures of TRAAK. In terms of the extracellular cap domain, TREK-1 and TRAAK show sequence similarity of 50.0%, which suggests they may have similar extracellular sites. Therefore, it is reasonable to use the crystal structure of TREK-1 to build the homology models of TRAAK mutants.

Comment 6: Investigation of ligand-binding mode of TRAAK using a chemical probe. English needs to be checked.

Line 224 28NH was tolerated by E38 and inserted deeply into the extracellular cap of TRAAK. What do the authors mean by ‘28NH was tolerated by E38’?

Line 240 Three independent molecular dynamics (MD) simulations were performed for the complex system S_{complex} for 500 ns, respectively. To investigate the influence of TKDC, an apo system S_{apo} without TKDC was established as a control under the same condition as S_{complex} , and applied to perform a 1- μ s long simulation. English needs revision.

Response: According to the reviewer’s comment, we have corrected the section title and the sentences,

and have polished the revised manuscript with a professional assistant in writing.

Comment 7: Line 243 TKDC interacted with I80, L102 and I106 at a high frequency (> 0.85). What is the meaning of >0.85 ?

Response: To characterize the binding mode of TKDC to TREK-1 in simulations, we calculate the frequency of interaction between a particular residue and TKDC. The “interaction” is defined based on distance: when a residue has one or more atoms presented within 0.4 nm of TKDC, it directly interacts with TKDC. In the last 200-ns trajectory of each S_{complex} system simulation, I80 and L102 directly interacted with the bicyclic group of TKDC at a high frequency (more than 0.85). We have revised the manuscript to include this information.

Comment 8: Line 247 During the last 200-ns simulations, distinct conformations were observed in the apo system S_{apo} and the complex system S_{complex} (Fig. 5). Conformations referred to the ligand or the channel? It is not clear.

Response: The conformations referred to the channel. We have revised the manuscript accordingly.

Comment 9: Line 276 This result also demonstrates that the extracellular cap may accommodate various inhibitors. What does this mean? At the same time? At the same location?

Response: No, this does not mean the extracellular cap may accommodate multiple inhibitors at the same location or at the same time. The discovery of new TREK-1 inhibitors with the different chemotypes demonstrates that the extracellular cap can be the binding target of the inhibitors belonging to the different chemotypes. We have revised the manuscript accordingly.

Comment 10: Line 293 Third, a chemical probe derived from TKDC was used to validate the proposed binding mode. Why TKDC was not used?

Response: Both TKDC and a chemical probe 28NH were used to validate the proposed extracellular binding sites of TREK-1, TREK-2 and TRAAK. However, the TKDC shows low inhibitory effect on TRAAK. 28NH was design by excluding the sulfonyl group of TKDC. Its potent inhibitory effect on TRAAK supports the proposed TRAAK model in which a glutamine located in the extracellular binding site negative contributes to the binding of TKDC.

Comment 11: Line 298 In this work, we identified the TKDC-binding site in the extracellular cap. However, the proposed binding site is only shown in the crystal structure of TREK-1 (Protein Data Bank (PDB) code 4TWK). Our mutagenesis study of TRAAK ruled out this possibility. Substitutions of the residues A35, E38 and V42 within the binding site in the extracellular cap of TRAAK

significantly increased the TKDC-induced. If the existence of a binding site in TREK-1 was already reported in the literature, what is the novelty of this study?

Response: The proposed binding site consisting of the E1 and E2 helices has not been reported in the literature.

Comment 12: Line 411 Ligand dockings of TKDC to TREK-1 were performed using both GLIDE software³¹ and RosettaLigand application. Why this was done only on TREK-1 and not the rest of channels considered? It is as if the studies are half-done. For some things one channel is used, for some others, the authors use other channels. It is fine if the choices are justified, but they are not.

Response: To investigate the possible binding modes of TKDC to TREK channels, we adopted a two-stage computational approach. In the first stage, molecular docking methods (Schrodinger Glide and Autodock) were applied to all TREK family crystal structures to identify possible binding site(s) of TKDC across the entire family. That is, Glide was applied to all of TREK-1, TREK-2 and TRAAK. In the second stage, considering the receptor flexibility and environmental effects, we applied RosettaLigand to accurately characterize the docking mode of TKDC into the identified binding site (site 3). Based on the original assay, TKDC had good potencies at both of TREK-1 and TREK-2. We have added RosettaLigand docking model of TREK-2/TKDC in the revised manuscript (please see the response to **Comment 4** for more information).

Comment 13: Line 414 Modeller was used to create homology models of human wildtype (WT) and three mutant TRAAK channels based on the crystal structure of TREK-1. Already I asked earlier about the reason behind this choice.

Response: Our mutagenesis study of TRAAK suggests three residues (A35, E38 and V42) contribute to binding to TKDC in the proposed binding site. However, this proposed site is only shown in a crystal structure of TREK-1 (PDB code 4TWK), but absent in the available crystal structures of TRAAK. In terms of the extracellular cap domain, TREK-1 and TRAAK show sequence similarity of 50.0%, suggesting they may have similar extracellular sites. Therefore, it is reasonable to use the crystal structure of TREK-1 to build the homology models of wild type and mutant TRAAK channels.

Comment 14: Line 427 filed with TIP3P - should be filled.

Response: Sorry for the mistake, we have had the manuscript polished with a professional assistant in writing.

Comment 15: Abstract: [...] And molecular dynamics simulations revealed the ligand-induced allosteric conformational transitions which obstructed the ion conductive path. Can this really be concluded from just one simulation?

Response: We performed three molecular dynamics simulations for the TKDC-binding complex system, and performed one simulation for the apo system without TKDC. According to the reviewer's comment, we performed one extra independent 1- μ s simulation for the apo system. The results from these two independent apo system simulations are highly consistent (supplementary Table 2). The average minimal distance between the bottom of E2 helix and the filter loop upstream pore helix P1 in the other subunit were 0.56 ± 0.17 nm and 0.54 ± 0.15 nm in these two simulations. This suggests that the helix E2 left clear extracellular ion conductive pathways in both simulations. All simulations suggest that the ligand-induced allosteric conformational transitions cause the blockage of the ion conductive pathway.

Additionally, an earlier cross-linking study of the other K2P channel TASK-3 by Clarke *et al.* [1] agrees with our assumption that the E2 helix can access to the pore region. By mutating one E70 residue at the bottom of E2 helix and one H98 residue in the pore region to cysteine in TASK-3, they find cross-linking can occur between the Cys-70 and Cys-98 residues on different subunits of the dimer. We have added this information in the discussion.

Supplementary Table 2 | Average minimal distance between the bottom of E2 helix and E1 helix in the simulation systems with (S_{complex}) and without TKDC (S_{apo}).

Simulation ID	Distance (nm)*
S_{apo} system simulation 1	0.71 ± 0.04
S_{apo} system simulation 2	0.73 ± 0.05
S_{complex} system simulation 1	0.96 ± 0.03
S_{complex} system simulation 2	0.82 ± 0.05
S_{complex} system simulation 3	0.87 ± 0.05

* indicates only backbone heavy atoms and last 200-ns trajectories were accounted in the calculation.

Reference:

[1] Clarke CE, Veale EL, Wyse K, Vandenberg JI, & Mathie A (2008) The M1P1 loop of TASK3 K2P channels apposes the selectivity filter and influences channel function. *The Journal of biological chemistry* 283(25):16985-16992.

Comment 16:

- Line 25 diseases. However, K2P channels are still lacking of the information of druggable site. However, information of the druggable sites in K2P channels is still lacking.
- Line 29 an allosteric ligand-binding site in the extracellular cap of the channels. ~~And~~ molecular dynamics
- Line 31 the ion conductive path. The presence of identified ligand-binding site in cap was confirmed by The identification of the ligand-binding site in the cap was confirmed by
- Line 33 that the extracellular caps of K2P channels can act as a new allosteric site and serve as direct drug

- Line 46 and ~~plays roles~~ is involved in glutamate conductance and the regulation of bloodbrain barrier
- Line 64 contribute to distinct traits in the way K2P channels interact with ligands. For example, the cap
- Line 68 against other ion channels. ~~However, how the cap responds to ligands is still unclear.~~
- Line 77 the extracellular cap of K2P channels is a functionally important ~~as a~~ drug targets.
- Line 159 (a) Binding site of TKDC in the extracellular cap of TREK-1 channel. (b) Detailed ~~show~~ view of

Response: Thank the reviewer for the instructive comment. We are sorry for the grammatical errors in our manuscript. We have revised manuscript with a professional assistant in writing.

Comment 17: Line 65 prevents direct ion transit between the pore mouth and the extracellular milieu, which explains the refractoriness of K2P channels to inorganic and toxin pore blockers^{22,23}. It is not very clear on what refractoriness means in this context. Maybe insensitivity? If so, I would change this word add something at the end like ‘that traditionally block K⁺ channels’

Response: Yes, K2P channels are insensitive to the classical potassium channel pore blockers. The extracellular domain of a K2P channel defines two tunnel-like side portals as extracellular ion pathway and partially obstructs the direct movement of ions into the extracellular milieu. Compared to the classical potassium channels, K2P channels offer bilateral extracellular access to the selectivity filter. This distinguishing extracellular ion pathway explains the insensitivity of K2P channels to the classical potassium channel pore blockers tetraethylammonium, 4-aminopyridine and caesium ion. According to the reviewer’s comment, we have revised the introduction section.

Comment 18: Line 136 To accurately dock TKDC to site 3, considering the receptor flexibility and environmental effects.. Just to make it clear that this was the only binding site to be analyzed further from the blind dock, if this interpretation is correct.

Response: Yes, it was only site 3 that was analyzed further from the blind dock. Please see the response to **Comment 12**.

Comment 19: Line 137 extracellular cap of TREK-1 was simulated using RossettaLigand application²⁴⁻²⁷. Is it necessary to say this here? This is also mentioned at least 3 times in the text.

Response: We have deleted the repeated descriptions in the manuscript.

Comment 20: Line 286 K2P channels represent important potential drug targets for various clinical treatments. Identifying ligand-binding sites in K2P channels is therefore of great importance. The utilization of chemical probes could facilitate the characterization of ligand-binding sites. This is really

vague and said in both abstract and intro already; is it necessary?

Response: According to the reviewer's comment, we have deleted these sentences in the discussion section.

Comment 21: Line 298 In this work, we identified the TKDC-binding site in the extracellular cap. However, the proposed binding site is only shown in the crystal structure of TREK-1 (Protein Data Bank (PDB) 300 code 4TWK), not in those of the other K2P channels, e.g., TRAAK^{14,15,17,18}. Does TKDC modulate TRAAK by the other domains rather than the extracellular cap?

Line 302 of TRAAK ruled out this possibility. Substitutions of the residues A35, E38 and V42 within the binding site in the extracellular cap of TRAAK significantly increased the TKDC-induced inhibition, indicating the proposed site is involved in modulating channels. Proteins are dynamic, and crystal structures represent some of their numerous dynamic conformations. The TKDC might insert into the extracellular cap of TRAAK in conformations different from the known crystal structures.

The mutagenesis data confirms that this site is important, but it does not really rule out that the ligand can bind elsewhere and have minor effects from other sites in TRAAK, as this has not been explored at all. Also, it is described very clearly the residual differences that cause the reduced sensitivity of TRAAK (and why) by both mutagenesis and docking in the results so it seems contradictory to just say now that the protein is dynamic and the binding site may present itself in other conformations of the protein. To me, it makes more sense to align the sequences of more K2P channels to see if the important residues are conserved to predict if any other channels may be selective to this ligand.

Response: L39 is a residue located in the proposed extracellular binding site of TRAAK. To validate that TKDC binds to the proposed extracellular binding site of TRAAK, we have tested the TKDC-induced inhibition of L39W TRAAK mutant. The basal activity of the channel is not altered by the mutation. The inhibitory effect of 100 μ M TKDC to the mutant is weaker than the TKDC-induced inhibition of the wild type TRAAK (Figure I). Both the loss-of-function mutant (L39W) and the gain-of-function mutants (A35Q, E38T and V42Q) indicate that the proposed extracellular ligand-binding site binds to TKDC. We acknowledge that we cannot completely rule out the possibility that TKDC binds to the other site in TRAAK. However, both mutagenesis and docking show the extracellular binding site is important in binding to the small molecule inhibitors of TRAAK. Because all crystal structures of TRAAK do not show this extracellular binding site, we propose that the TKDC may insert into the extracellular cap of TRAAK in conformations different from the known crystal structures. According to the reviewer's comment, we have revised the manuscript.

Figure I. Inhibitions of wide type and L39W mutant TRAAK by 100 μ M TKDC.

Unpaired t-test was used for statistical analysis. The data are shown as mean \pm SEM.

Comment 22: Line 309 As an inhibitor targeting TREK-1, TKDC exhibited obvious antidepressant-like effects (supplementary Fig. 9), which is consistent with previous behavioral tests indicating an antidepressant effect when knocking out the *Trek1* gene in mice^{28,29}. Maybe I do not understand the experimental results, but these experiments do not seem to be mentioned in the results.

Response: We have moved the experimental data into the Results section.

Comment 23: Line 312 behavior tests performed in this work, the antidepressant-like effects of TKDC were observed and compared to a known antidepressant drug, fluoxetine. Well-known as an antagonist of the 5-HT_{2B} receptor, fluoxetine can also inhibit TREK channels^{6,13}. I understand that they authors have compared it to fluoxetine because it is a known inhibitor, but it does not seem to add any extra insight as they make it very clear it is very different in terms of location and sequence. I would prefer a discussion of how this site apparently physically blocks the channels via the cap, as this seems more relevant to me and unique to K2P channels.

Response: Our MD simulations suggest that the E2 helix can access to the pore region and block the extracellular ion pathway. An earlier cross-linking study of the other K2P channel TAKS-3 by Clarke *et al.* [1] agrees with this proposed inhibition mechanism. By mutating one E70 residue at the bottom of E2 helix and one H98 residue in the pore region to cysteine in TASK-3, they find cross-linking can occur between the Cys-70 and Cys-98 residues on different subunits of the dimer. Since the bottom of the E2 helix can access to the pore region, it is possible for this helix to physically block the extracellular ion pathway, which begins at the outer mouth of the pore. According to the reviewer's comment, we have added this information in the discussion section.

Reference:

[1] Clarke CE, Veale EL, Wyse K, Vandenberg JI, & Mathie A (2008) The M1P1 loop of TASK3 K2P channels apposes the selectivity filter and influences channel function. *The Journal of biological*

Reviewer #3 (Remarks to the Author):

This is an interesting study that uses a combination of electrophysiological profiling, chemistry, mutagenesis, molecular modeling and virtual screening to identify novel classes of potentially selective (and in vivo active) modulators of the TREK subfamily of K2P channels. Overall, the findings are interesting and the combination of approaches largely complementary in supporting the authors' main conclusions.

Comment 1: The initial description of the “fortuitous” identification of TKDC from an internal library of ~1000 molecules needs a little more elaboration. What library are the authors referring to? Presumably an in-house library that has been developed over time, or was it curated in a particular manner etc?

Response: The in-house library was developed over time. It includes ~1000 small molecules purchased or synthesized for our previous and current research projects of drug discovery. According to the reviewer's comment, we have revised the manuscript and have added more information in the Methods.

Comment 2: Determination of inhibitor potencies and statistical comparisons. How was the potency for TKDC established at TRAAK, given that it had very small effects up to the highest concentrations tested? (Fig. 1d). The authors have clearly assumed a complete inhibition to be able to derive an IC50 value, but this should be explicitly stated as a caveat. More importantly, all potency estimates are given as IC50- +/- SEM, but it is well established that drug potency (or affinity) values are only Gaussian (or approximately so) when expressed as logarithms. Thus, ALL the statistics shown in the manuscript are actually invalid (t tests and ANOVAS assume a Gaussian distribution), unless they have actually been performed on the logarithms of the potency values. This needs to be clarified and corrected.

Response: Because TKDC is not very soluble, we cannot conduct the experiment and observe a complete inhibition of TRAAK at a very high concentration of TKDC. But we observed that ~75% decrease in the current when 300 μM TKDC was perfused in the whole-cell voltage clamp experiments of TRAAK. We have performed more independent whole-cell voltage clamp experiments for TRAAK with TKDC at different concentrations (Table I).

We have fitted all previous and new data using doseResp function in OriginPro 8.1 software (Fig. 1d). The half inhibition concentrations were derived from fits of the dose-response curves to the function:

$$\frac{I}{I_0} = A_1 + \frac{A_2 - A_1}{1 + 10^{(\log x_0 - x)P}}$$

Where I_0 and I are current amplitudes before and after application of inhibitors; A_1 and A_2 are constants between 0 and 1; x is the concentration of inhibitor; x_0 is the concentration when 50% inhibition was reached (IC₅₀), P is the Hill constant.

In statistical analyses, the experimental data were expressed as mean ± SEM. Origin 8.1 software

(OriginLab Corporation, Northampton, USA) was used in the analyses of the cellular experiments. The averaged IC_{50} was collected from the fitting results of each patch (lasting for five inhibitor concentrations), and then was used to compare between different groups. Unless specified, an unpaired t test was used to compare two means and a one-way ANOVA along with post-hoc LSD test was used to compare three or more mean. When the data distribution was skewed, the independent-sample Kruskal-Wallis test (non parametric test) followed by Dunn-Bonferroni post hoc test was performed. The results of animal behavioral tests (forced swimming test, tail suspension test and open field test) were analyzed using one-way ANOVA with post-hoc LSD tests in SPSS software. $P < 0.05$ was considered statistically significant. We have revised the manuscript to include this information in the Methods.

Table I | Dose-dependent inhibition of TRAAK by TKDC.

Concentration of TKDC (μM)	Normalized inhibition I/I_{control} (%)
1	0.97 ± 0.01
3	0.93 ± 0.03
10	0.80 ± 0.05
30	0.65 ± 0.06
100	0.45 ± 0.05
200	0.31 ± 0.06
300	0.25 ± 0.01

Figure 1. Inhibition of TREK subfamily channels by TKDC in CHO cells.

(a) Chemical structure of TKDC. (b, c) Typical whole-cell current traces recorded from CHO cells overexpressing the TREK-1 channel with 10 μM TKDC (b) or DMSO application (c). Currents were elicited by depolarizing voltage steps from a holding potential of -80 mV to +80 mV in 20 mV increments and then stepping down to -60 mV. (d) Dose-dependent inhibition of TKDC on TREK-1, TREK-2 and TRAAK channels. (e) The statistics of the half-inhibitory concentrations of TKDC to TREK-1 ($n = 7$), TREK-2 ($n = 7$), and TRAAK ($n = 8$) channels. IC_{50} values were obtained via

dose-response fitting. Kruskal-wallis test was used for statistical analysis; ** indicates $P < 0.01$. The data are shown as mean \pm SEM.

Comment 3: In the first description of the modeling results in the main manuscript, when discussing the proposed binding mode of TKDC to TREK-1, a little more elaboration is required for the reader to be able to better understand. Rather than simply stating “Using the molecular docking method...”, The authors need to be more explicit. Based on my reading of the Methods, the authors actually adopted a two-stage approach, is that correct? In the first stage, Schrodinger GLIDE using default parameters was applied to the TREK family crystal structures to identify a variety of possible poses across the entire family. Based purely on the docking scores from these studies (< -8.0), the seven highest ranked poses were chosen. The second stage of the modeling involved both GLIDE and RosettaLigand to more accurately understand the proposed docking of TKDC into TREK-1 (in what was essentially “Site 3” from the first stage of modeling). This two-stage approach should be explained better in the Results.

Response: Yes, we adopted a two-stage approach. In the first stage, molecular docking methods (Schrodinger Glide and Autodock) were applied to the TREK family crystal structures to identify possible binding site(s) of TKDC across the entire family. In the second stage, considering the receptor flexibility and environmental effects, RosettaLigand was applied to accurately characterize the docking mode of TKDC into the identified binding site. According to the reviewer’s suggestion, we have revised the manuscript accordingly.

Comment 4: While still on the modeling, I find it quite interesting that 5 of the 7 highest ranked poses were from the TRAAK structures, despite the fact that TKDC had lowest potency for this receptor! The authors make a reasonable case for ruling out the poses and settling on Site 3 but what would happen, for instance, if they looked at Site 9, which was also in TREK-1, docked at a lower site than Site 3 and with a slightly lower score? I guess my main concern here is the rather arbitrary nature in using initial docking scores (especially since most of the best scores come from the “wrong” channel) to choose the right one. Of course, much of this concern is mitigated by many of the subsequent experiments performed by the authors, but it is surprising that the two receptors for which TKDC shows highest potency, TREK-1 and TREK-2, yielded that fewest high ranked poses. I discuss some questions regarding TREK-2 below.

Response: Thank the reviewer to point out that the binding sites predicted by the docking methods were not described clearly. In fact, some of the 36 predicted binding sites were equivalent sites in the different crystal structures. For example, all of sites 1, 2, 4, 5, 6, 7, 8, 11 and 21 were binding pockets consisting of the M1, M2, M4 and P1 helices and shared conservative residues. According to their locations and compositions of residues, we assigned all predicated binding sites into eleven groups (group A-K). All sites were ranked based on the Glide G-score and Autodock binding score, respectively. Regardless of docking methods (Glide and Autodock), the predicated binding sites in groups A and B always had the lowest scores (Glide G-scores < -7.5 or Autodock binding score < -9.0) (supplementary Fig. 3b). In the sites of group A (sites 1, 2, 4, 5, 6, 7, 8, 11 and 21), TKDC was surrounded by hydrophobic residues, which provided an unfavorable environment for the charged

sulfonyl group of TKDC (supplementary Fig. 4a-f). Therefore, the sites of group A may not bind to TKDC. In group B, there was only one member, i.e., site 3, which had the third lowest G-score and the lowest Autodock binding score. Site 3 was located in the extracellular cap domain consisting of two helices from different subunits in TREK-1. As an extracellular groove observed only in a crystal structure of TREK-1 (PDB code 4TWK), it has not been reported to bind inhibitors of K2P channels. In site 3, TKDC showed a more reasonable pose, in which its hydrophobic groups were inserted into a gap between the extracellular helices and the polar sulfonyl group was exposed to the extracellular solvent (supplementary Fig. 4g-h). Thus, site 3 was considered as the possible binding site of TKDC in TREK-1 and was selected for further analysis. We have rewritten this part in the section of “binding mode of TKDC to TREK-1”.

Supplementary Figure 3 | Docking of TKDC to TREK channels

(a) Potential ligand-binding sites in the crystal structures of TREK channels. Each predicated binding site is indicated as a sphere. (b) Glide G-scores and Autodock binding scores of docking TKDC to each potential binding site. The binding sites were assigned in the different groups according to their locations and compositions of residues. Eleven different groups are shown in different colors.

Supplementary Figure 4 | Docking models of TKDC in the binding sites of groups A and B generated using Glide and Autodock

(a-f) Representative docking poses of TKDC in the sites of group A, including (a, b) sites 1, 4, 5, 6, 7 and 8, (c, d) sites 11 and 21, and (e, f) site 2. (g, h) Representative docking poses of TKDC in the site 3 of group B. These models were generated using (a, c, e, g) Glide and (b, d, f, h) Autodock. TKDC and the protein residues in the binding sites are shown as sticks. The hydrophobic residues interacting with the charged sulfonyl group of TKDC are highlight in yellow. Residues blocking view are omitted.

Comment 5: The mutational validation of the EC cap in TREK-1 seems convincingly done, but I am

surprised as to why the authors did not mutate T79. Mutation of this residue should be tested, given that all the other residues were tested (including Q83 and Q105, which had no effect upon alanine substitution).

Response: T79A mutants showed no significant changes in TKDC-induced inhibition compared with WT TREK-1 channel (supplementary Table 1). We have added this information in the revised manuscript.

Supplementary Table 1 | Inhibitory effects of TKDC on mutant TREK-1 channels T76A, Q83A, and Q105A.

Construct	IC ₅₀ (μM)	Number of recorded cells
T76A TREK-1	2.4 ± 0.4	6
Q83A TREK-1	2.4 ± 0.6	4
Q105A TREK-1	2.2 ± 0.5	4

Comment 6: The introduction of key TREK-1 residues into TRAAK (Figure 3) provides good support for the role of the extracellular cap in inhibitor action. However, the resulting curves appear distinctly biphasic to me. Does this mean that the modulator can actually bind to more than one site? This is of relevance due to one of the points the authors themselves make in the Discussion.

Response: The insufficient data resulted in the biphasic curves. We have performed more independent whole-cell voltage clamp experiments for the wild type and mutant TRAAK channels with TKDC. As shown in Fig. 3b, the resulting curves are not biphasic. We have revised the figure in the manuscript.

Figure 3. Substitutions of key residues at extracellular cap exhibited enhanced TKDC-induced inhibition of TRAAK channel.

(a) Multiple sequence alignment for the extracellular caps of TREK-1, TREK-2, and TRAAK channels. (b) Dose-dependent inhibition of TKDC on the WT and mutant TRAAK channels. (c) Histograms summarizing the half-inhibitory concentrations of TKDC on the WT ($n = 8$), A35Q ($n = 8$), E38T ($n = 6$), and V42Q ($n = 7$) mutant TRAAK channels. Mutations of these residues in the extracellular cap showed enhanced inhibitory effects of TKDC. IC_{50} values were obtained via dose-response fitting. One-way ANOVA with post-hoc LSD test was used for statistical analysis [$F(3, 20) = 18.551$]; ** indicates $P < 0.01$. The data are shown as mean \pm SEM. (d-e) Docking modes of TKDC in A35Q TRAAK (d), in V42Q TRAAK (e), and in E38T TRAAK (f). The protein is shown as a cartoon. TKDC and key residues are shown as sticks.

Comment 7: With regards to the MD simulations (Fig. 5), are there any existing data from other studies of this channel class (e.g., mutagenesis or cross-linking, for instance) to support the possibility of an allosteric transition of E2 similar to that being proposed to be caused by the binding of TKDC? Also, the authors are highlighting the final pose at the end of the MD run, but what does the starting pose look like? They discuss interaction frequencies, but it may also help visually to at least see how the compound started relative to how it ends to further highlight stability.

Response: Thank the reviewer for the instructive suggestions. An earlier cross-linking study of the other K2P channel TAKS-3 by Clarke *et al.* suggests that E70 residue at the bottom of E2 helix and

H98 residue in the pore is close to each other [1]. By mutating one E70 residue and one H98 residue to cysteine in TASK-3, they find cross-linking can occur between the Cys-70 and Cys-98 residues on different subunits of the dimer. Since the bottom of the E2 helix can access to the pore region, it is possible for this helix to physically block the extracellular ion pathway. According to the reviewer's comment, we have added this information in the Discussion.

The TKDC-binding complex system was built by inserting the docking model of TREK-1/TKDC in the membrane environment. Thus, the starting pose of TKDC is the one shown in the docking model (Figure 2a-b in the manuscript). In simulations, the dominant pose of TKDC (supplementary Fig. 7 in the manuscript) was very similar to its starting pose.

Reference:

[1] Clarke CE, Veale EL, Wyse K, Vandenberg JI, & Mathie A (2008) The M1P1 loop of TASK3 K2P channels apposes the selectivity filter and influences channel function. *The Journal of biological chemistry* 283(25):16985-16992.

Comment 8: The use of a virtual screen to identify additional classes of inhibitor is an additional strength of this MS. However, I am struck by the fact that the authors did not do a fairly obvious experiment, which is to test them (at least) on TREK-2 and TRAAK. This also speaks to the issue of the potential for selectivity vs off-target activity. I accept that the latter may be asking too much, but the acknowledgment of the potential for other target effects should be addressed, at least in future studies. However, I do feel that selectivity testing across TREK-2 and TRAAK is required.

Response: We have tested TKN1 and TKN2 on TREK-2 and TRAAK, respectively. Both of TKN1 and TKN2 inhibit TREK-2 and TRAAK (supplementary Fig. 9 and supplementary Table 3). Compared with TRAAK, TREK-2 is more sensitive to these two inhibitors. We have added this information in the revised manuscript.

Supplementary Figure 9 | Dose-dependent inhibition of TREK-2 (a) and TRAAK (b) by TKN1 and TKN2.

IC_{50} values were obtained by dose-response fitting.

Supplementary Table 3 | Inhibitory effects of TKN1 and TKN2 on TREK-2 and TRAAK channels.

Construct	Inhibitor	IC ₅₀ (μM)	Number of recorded cells
TREK-2	TKN1	4.4 ± 1.0	6
TREK-2	TKN2	1.7 ± 0.8	3
TRAAK	TKN1	15.7 ± 3.1	6
TRAAK	TKN2	10.3 ± 1.5	6

Comment 9: Finally, I may have missed this, but I am surprised to see a lack of docking of any of the inhibitors, at least TKDC, into TREK-2. Based on the original assay, this had as good, if not better, potency at TREK-2 than TREK-1, yet the latter is used as the template for all subsequent studies due to the identification of “Site 3”. At the very least the authors need to be able to explain the effect on TREK-2 in terms of their proposed model, or else need to suggest an alternative site?

Response: According to the reviewer’s comment, we have revised the manuscript and added RossettaLigand docking model of TREK-2/TKDC. According to the multiple sequence alignment of the extracellular caps of TREK channels, TREK-1 and TREK-2 show a sequence similarity of 66.7%. This suggests TREK-2 probably have extracellular TKDC-binding sites similar to the site 3 of TREK-1, although those sites were absent in its available crystal structures. We used the TREK-1 crystal structure as a template to build the homology model of TREK-2 and performed RossettaLigand docking of TKDC to the proposed extracellular binding site. The docking model of TREK-2/TKDC with the lowest binding energy (supplementary Fig. 5) was very similar to the one of TREK-1/TKDC. In the binding site of TREK-2, the bicyclic group of TKDC was fully enveloped by a cavity formed by residues I105, Q108, L127 and H130. Close to the bottom of the extracellular cap domain, the chlorophenyl group of TKDC was inserted in a cavity formed by Q101, T104 and I105. The negatively charge sulfonyl group of TKDC was attracted by the adjacent positively charged H130, which might stabilize the binding of TKDC to this site.

Supplementary Figure 5 | Binding mode of TKDC to TREK-2

TKDC and the residues in the extracellular binding site are shown as sticks. The E1 and E2 helices are shown as green and cyan cartoons.

Reviewer #4 (Remarks to the Author):

K2P family channels control cellular electrical excitability in a wide range of physiological contexts. They generate background K⁺ currents to set and maintain the resting membrane potential and are additionally regulated by a diverse set of modulators. Pharmacological modulation of their activity would provide a means to dissect the biological roles of K2Ps and has potential for clinical applications in numerous contexts. However, development of specific pharmacology has been difficult and remains an important goal. In this manuscript, Chen, Cheng, Luo et. al identify a new small-molecule blocker of TREK subtype K2Ps called TKDC. They use a combination of electrophysiological and computational approaches to present a model for block by TKDC through binding to a unique extracellular cap domain in the channel. In addition, they provide evidence from behavioral studies consistent with its action on TREK channels in vivo. This work is overall nicely done and very interesting. I offer below some suggestions for consideration that I believe would improve the manuscript and make the findings immediately more useful for the field:

Comment 1: Add an experiment testing block of channel by TKDC of TREK-1 or -2 in inside-out vs. outside-out patches. This would provide additional evidence that block is truly direct and could demonstrate that inhibition strictly requires access to the extracellular cap domain as predicted by the model presented.

Response: Thank the reviewer for the valuable advice. We have performed experiments testing block of the TREK-1 channel by TKDC in inside-out and outside-out patches. The application of TKDC to the extracellular face of the membrane (outside-out recording) consistently reduced the open probability of TREK-1 (supplementary Fig. 2a). In contrast, the application of TKDC to the intracellular side of the channel did not significantly inhibit TREK-1 activity (supplementary Fig. 2b). These results provide evidence that the TKDC directly inhibits TREK-1 and the inhibition requires access to the extracellular cap domain. We have revised the manuscript to include this information.

Supplementary Figure 2 | Inhibition of TKDC on the opening of TREK-1.

(a, b) Inhibitory effect of 10 μM and 30 μM TKDC on TREK-1 open probability in (a) outside-out and (b) inside-out configuration. Traces were obtained with pipette voltage of +80 mV. The bold bar above the current trace shows the application and washout of TKDC.

Comment 2: Does drug binding affect activation of the channels in addition to basal activity (i.e. by

mechanical force, lipids, or pH change)? A comprehensive survey would be nice, but even evaluating block +/- one activating stimulus would be a very useful piece of information to know for those interested in using this drug.

Response: We have tested the TKDC-induced inhibition of TREK-1 in addition to the mechanical force using outside-out patch. 10 μ M TKDC was applied via dish perfusion. The bath level was kept as low as possible to reduce noise. Currents were recorded at a pipette potential of +80 mV. Mechanical stimulation (-7 mmHg) for patches was applied by suction for the outside-out configuration using a pressure monitor (PM 015D, World Precision Instruments) controlled by clampex 10.6 software. As shown in Figure II, TKDC suppressed the activation of TREK-1 by mechanical force.

The main aim of this work is to reveal an allosteric ligand-binding site in the extracellular cap of K2P channels. We will comprehensively study the effects of drug binding on the activation of the TREK channels (by mechanical force, lipids, or pH change) in the future work.

Figure II | Effect of TKDC on the activation of TREK-1 by mechanical force

Inhibitory effect of 10 μ M TKDC on TKDC activated by mechanical force in the outside-out configuration. Traces were obtained with pipette voltage of +80 mV. The bold bar above the current trace shows the application of TKDC.

Comment 3: Does TKDC block other K2Ps? Testing block of at least one K2P from each clade of the family would accomplish two things: it would be a complementary test of the mechanism proposed and it would make the drug immediately more useful to the field if its specificity among the entire K2P family is characterized.

Response: In this study, we mainly focused on the identification of the extracellular ligand-binding site of TREK channels. In the future work, we will further investigate the specificity of the inhibitors of TREK channels among the entire K2P family and will discover TRAAK specific ligands.

Comment 4: Both binding and washout of TKDC appear slow in electrophysiological recordings (Fig. S1D). A discussion of why this is would be interesting. Is it consistent with the molecular dynamics simulations/crystal structures and indicative of an infrequently populated conformation that exposes the binding site to solution for drug access?

Response: To keep stable whole-cell recordings, the rate of solution perfusion is slow, which results in

the time scale in the figure. Both binding and dissociation (washout) of TKDC are not slow.

Comment 5: The presentation of the antidepressant-like effects of TKDC in vivo should be moved out of the discussion and into the main text.

Response: We have moved the experimental data into the main text.

Comment 6: The statistical analysis used are appropriate.

Response: We fitted data using doseResp function in OriginPro 8.1 software. The half inhibition concentrations were derived from fits of the dose-response curves to the function:

$$\frac{I}{I_0} = A_1 + \frac{A_2 - A_1}{1 + 10^{(\log x_0 - x)P}}$$

Where I_0 and I are current amplitudes before and after application of inhibitors; A_1 and A_2 are constants between 0 and 1; x is the concentration of inhibitor; x_0 is the concentration when 50% inhibition was reached (IC_{50}), P is the Hill constant.

In statistical analyses, the experimental data were expressed as mean \pm SEM. Origin 8.1 software (OriginLab Corporation, Northampton, USA) was used in the analyses of the cellular experiments. The averaged IC_{50} was collected from the fitting results of each patch (lasting for five inhibitor concentrations), and then was used to compare between different groups. Unless specified, an unpaired t test was used to compare two means and a one-way ANOVA along with post-hoc LSD test was used to compare three or more mean. When the data distribution was skewed, the independent-sample Kruskal-Wallis test (non parametric test) followed by Dunn-Bonferroni post hoc test was performed. The results of animal behavioral tests (forced swimming test, tail suspension test and open field test) were analyzed using one-way ANOVA with post-hoc LSD tests in SPSS software. $P < 0.05$ was considered statistically significant. We have revised the manuscript to include this information in the Methods.

REVIEWERS' COMMENTS:

Reviewer #1 (Remarks to the Author):

Authors have addressed most of my concerns. However, there are still a number of awkward sentences in the revised text. Authors should go over carefully and try to correct them as much as possible. Some of these are:

"Therefore, these extracellular domains may play promising roles in the discovery of specific drugs of K2P channels against other ion channels."

'These results suggest that TKDC is a common inhibitor for all three members of TREK channels with different subtype potencies.'

"As an extracellular groove observed only in a crystal structure of TREK-1 (Protein Data Bank (PDB) code 4TWK), it has not been reported to bind to the inhibitors of the K2P channels."

"reasonable pose with"? "binding pose"?

"Depressive-related behavior tests"? Makes no sense.

"three or more mean." There are others.

Too many sentences about "discovering new drugs based on the findings". Once is enough.

Reviewer #2 (Remarks to the Author):

I am happy with the replies provided by the authors to my original lengthy report.

Reviewer #3 (Remarks to the Author):

The authors have adequately addressed all my comments. The additional experiments performed have strengthened the manuscript. My only additional comment is that the new Supp Table 1 has mis-labeled mutation T79A as "T76A"! This should be corrected.

Reviewer #4 (Remarks to the Author):

The authors have satisfactorily addressed my comments and I recommend that the manuscript be published.

Responses to the comments of reviewers

Reviewer #1:

Authors have addressed most of my concerns. However, there are still a number of awkward sentences in the revised text. Authors should go over carefully and try to correct them as much as possible. Some of these are:

“Therefore, these extracellular domains may play promising roles in the discovery of specific drugs of K2P channels against other ion channels.”

‘These results suggest that TKDC is a common inhibitor for all three members of TREK channels with different subtype potencies.’

“As an extracellular groove observed only in a crystal structure of TREK-1 (Protein Data Bank (PDB) code 4TWK), it has not been reported to bind to the inhibitors of the K2P channels.”

“reasonable pose with”? “binding pose”?

“Depressive-related behavior tests”? Makes no sense.

“three or more mean.” There are others.

Too many sentences about “discovering new drugs based on the findings”. Once is enough.

Response: We have used the English language editing service of Nature Publishing Group Language Editing and ensured this manuscript is clear.

Reviewer #2:

I am happy with the replies provided by the authors to my original lengthy report.

Reviewer #3:

The authors have adequately addressed all my comments. The additional experiments performed have strengthened the manuscript. My only additional comment is that the new Supp Table 1 has mis-labeled mutation T79A as "T76A"! This should be corrected.

Response: We have revised the Supplementary Table 1 accordingly.

Reviewer #4:

The authors have satisfactorily addressed my comments and I recommend that the manuscript be published.